# SARS-CoV-2 and Its Bacterial Co- or Super-Infections Synergize to Trigger COVID-19 Autoimmune Cardiopathies

**DOI:** 10.3390/ijms241512177

**Published:** 2023-07-29

**Authors:** Robert Root-Bernstein, Jack Huber, Alison Ziehl, Maja Pietrowicz

**Affiliations:** Department of Physiology, Michigan State University, East Lansing, MI 48824, USA; huberja2@msu.edu (J.H.); ziehlali@msu.edu (A.Z.); pietro10@msu.edu (M.P.)

**Keywords:** COVID-19, SARS-CoV-2, myocarditis, endocarditis, MIS-C, autoimmune, cross-reactive, molecular mimicry, antigenic complementarity, cardiolipin, myosin, laminin, *Streptococci*, *Staphyococci*, *Enterococci*, *Klebsiella*, adenovirus, vaccine

## Abstract

Autoimmune cardiopathies (AC) following COVID-19 and vaccination against SARS-CoV-2 occur at significant rates but are of unknown etiology. This study investigated the possible roles of viral and bacterial mimicry, as well as viral-bacterial co-infections, as possible inducers of COVID-19 AC using proteomic methods and enzyme-linked immunoadsorption assays. BLAST and LALIGN results of this study demonstrate that SARS-CoV-2 shares a significantly greater number of high quality similarities to some cardiac protein compared with other viruses; that bacteria such as *Streptococci*, *Staphylococci* and *Enterococci* also display very significant similarities to cardiac proteins but to a different set than SARS-CoV-2; that the importance of these similarities is largely validated by ELISA experiments demonstrating that polyclonal antibodies against SARS-CoV-2 and COVID-19-associated bacteria recognize cardiac proteins with high affinity; that to account for the range of cardiac proteins targeted by autoantibodies in COVID-19-associated autoimmune myocarditis, both viral and bacterial triggers are probably required; that the targets of the viral and bacterial antibodies are often molecularly complementary antigens such as actin and myosin, laminin and collagen, or creatine kinase and pyruvate kinase, that are known to bind to each other; and that the corresponding viral and bacterial antibodies recognizing these complementary antigens also bind to each other with high affinity as if they have an idiotype-anti-idiotype relationship. These results suggest that AC results from SARS-CoV-2 infections or vaccination complicated by bacterial infections. Vaccination against some of these bacterial infections, such as *Streptococci* and *Haemophilus*, may therefore decrease AC risk, as may the appropriate and timely use of antibiotics among COVID-19 patients and careful screening of vaccinees for signs of infection such as fever, diarrhea, infected wounds, gum disease, etc.

## 1. Introduction

Autoimmune complications sometimes follow infection with SARS-CoV-2, the virus that causes COVID-19 [1,2,3], or, more rarely, following vaccination against SARS-CoV-2 [4,5,6,7]. Myocarditis or pericarditis are the most common autoimmune complications [1,2,3,4,5,6,7]. Estimates of the prevalence of myocarditis among COVID-19 patients range from a low of 20–30% of hospitalized patients [8,9] to a high of 50–70% among intensive care unit (ICU) patients and post-recovery COVID-19 patients [9,10,11] and from a low of 1–2% among mild cases to 10–20% [9,10]. Myocardial injury as measured by increased troponin significantly increased the probability of death among hospitalized COVID-19 patients (51.2% vs. 4.5%; *p* < 0.001), correlated significantly with a diagnosis of acute respiratory distress syndrome (ARDS) (58.5% vs. 14.7%), and predicted the need for invasive or non-invasive respiratory support (46.3% vs. 3.9%) [9,12,13,14]. Most importantly, from the perspective of autoimmunity, a significant proportion of myocarditis patients have been polymerase chain reaction (PCR) negative but positive for SARS-CoV-2 antibodies, suggesting that their cardiac complications have followed the resolution of their viral infection [10,12,15,16,17]. In sum, SARS-CoV-2-infected individuals were 18.28 times more likely to develop myocarditis than uninfected, unvaccinated individuals [18], with hospitalized COVID-19 patients having a 20–60% probability of developing myocarditis and as many as 70% of COVID-19 survivors displaying clinical signs of myocardial damage more than a month after their recovery.

Myocarditis has also been reported rarely among SARS-CoV-2 vaccinees. A study of clinically apparent myocarditis among U.S. military personnel vaccinated with either the BNT162b2-mRNA vaccine (Pfizer-BioNTech) or the mRNA-1273 vaccine (Moderna) found a rate of about 1 in 100,000 vaccinees [19], a rate about 3 times lower than reported to the United States Vaccine Adverse Event Reporting System (VAERS) [20] and much less than the 5/100,000 reported among Israeli military personnel [21]. Other studies suggest an incidence rate as high as 10 to 20 cases per 100,000 subjects [7,22,23,24]. The majority of cases are among men under the age of 65 [4,6].

The data summarized above poses a number of puzzles. One is why the vast majority of people who contract COVID-19 do not develop autoimmune cardiopathies; however, the incidence of these complications becomes increasingly likely the more serious an individual’s COVID-19 disease becomes, so that they affect the majority of intensive care unit patients. Similarly, what explains the much lower incidence of post-vaccinal myocarditis and its increased incidence among young men as compared with other COVID-19 vaccines? In this context, it is important to realize that one of the differences between people getting vaccinated against COVID-19 and those being hospitalized for it or admitted to intensive care is their probability of having a bacterial co-infection. This risk is presumably extremely low among vaccinees, who are likely to be healthy, but has been estimated to be between 30% and 90% among hospitalized COVID-19 patients [25,26,27,28,29,30,31,32]. *Streptococci*, *Staphylococci*, *Klebsiella*, *E. coli,* and *Enterococcus faecium* are the most commonly diagnosed SARS-CoV-2 co- or super-infections [25,26,27,28,29,30,31,32]. These bacteria are, notably, among the bacteria most often associated with triggering endocarditis both prior to and during the COVID-19 pandemic [33,34,35,36] and therefore possible triggers for cardiac autoimmunity, and their presence preceding some cases of COVID-19-related autoimmune myocarditis has been documented [37,38]. Myocarditis following COVID-19 vaccination has been suggested to result from accidental intravenous inoculation of the vaccine, which would explain the rarity with which autoimmune complications are observed [39] but not the unusual risk in young men.

The autoantigen targets of autoimmune myocarditis have been well-characterized in human patients and include, but are not limited to, actin, beta-adrenergic, cardiolipin, collagen, receptors, creatine kinase, alpha enolase, laminin, myosin, nuclear antigens, pyruvate kinase, tropomyosin, and troponin [36,40,41,42,43,44,45]. Autoantibodies against many of these proteins have been documented among hospitalized COVID-19 patients (Table 1) [46,47,48,49,50,51,52,53]. The mechanism by which autoantibodies against these antigens are elicited, however, remains obscure. Among the mechanisms that have been proposed are intermittent virus shedding, viral reactivation, or reinfection with another SARS-CoV-2 strain [54], molecular mimicry between SARS-CoV-2 and cardiac antigens [2], and molecular mimicry enhanced by a bystander infection promoting a hyperinflammatory environment [3]. Direct ACE-2-mediated cardiomyocyte damage by the virus; microvascular disease with vascular leakage and hypercoagulation due to endothelitis; systemic hyperacute inflammatory response syndrome; and pneumonia-related oxygen supply-demand imbalance with ischemia [11]. Oddly, given the significant literature linking myocarditis and rheumatic heart disease to viruses and bacteria that display molecular mimicry to heart proteins [41,55,56,57], this autoimmune disease mechanism has been proposed [4,5,6,7] but not investigated with regard to COVID-19-associated autoimmune cardiopathies, nor has the possibility that bacteria are involved in the pathogenesis of this autoimmunity been explored. The purpose of this paper is to test whether SARS-CoV-2 antibodies mimic cardiac antigens and whether such mimicry is sufficient to explain the risks of autoimmune myocarditis associated with COVID-19 and vaccination against it. The possible role of bacterial co-infections, either as bystander infections or active triggers of autoimmune myocarditis, is also explored.

## 2. Results

As a first step in identifying possible triggers of myocardial autoimmunity following SARS-CoV-2 infections, two types of similarity searches were carried out. BLAST was used to identify similarities between human proteins and bacterial or viral proteomes, while LALIGN was used to identify similarities between specific pairs of human proteins and SARS-CoV-2 proteins. Only matches that had a Waterman-Eggert score of at least 50, an E value of less than 1.0, and contained a sequence of ten amino acids in which at least six were identical were counted as sufficiently similar to induce possible cross-reactive immunity; this criterion is based on substantial research demonstrating that sequences exhibiting at least this degree of similarity have a >85% probability of being cross-reactive under experimental conditions [58,59,60,61].

BLAST and LALIGN results of this study demonstrate that SARS-CoV-2 shares a significantly greater number of similarities to some cardiac proteins and that these similarities are of higher quality compared with other viruses; that bacteria such as *Streptococci* and *Staphylococci* also display very significant similarities to cardiac proteins but to a different set than SARS-CoV-2; that these similarities are largely validated by ELISA experiments demonstrating that polyclonal antibodies against SARS-CoV-2 and bacteria that are often found as co- or super-infections in COVID-19 recognize cardiac proteins with high affinity; that to account for the range of cardiac proteins targeted by autoantibodies in COVID-19-associated autoimmune myocarditis, both viral and bacterial triggers are probably required; that the targets of the viral and bacterial antibodies are often molecularly complementary antigens such as actin and myosin that are known to bind to each other; and that the corresponding viral and bacterial antibodies recognizing these complementary antigens also bind to each other with high affinity as if they have an idiotype-anti-idiotype relationship. Each of these results will be presented in a separate section.

### 2.1. Proteomic Study of Similarities Shared by SARS-CoV-2 and Human Cardiac Proteins

Proteomic studies of virus-cardiac protein similarities were carried out using LALIGN, resulting in pairwise comparisons summarized in Figure 1. This method allowed the potential contribution of each to COVID-19-associated cardiomyopathies to be predicted. A special focus was put on the spike protein of SARS-CoV-2, since that is used in most COVID-19 vaccines and is the most likely trigger for associated post-vaccinal AM. The similarities for all the SARS-CoV-2 proteins were summed for each cardiac protein to obtain the total number of similarities contained in the virus as a whole. Some of the most noteworthy similarities between human cardiac proteins and those of SARS-CoV-2 are provided below in Figure 2 (SARS-CoV-2 spike protein) and Figure 3 (SARS-CoV-2 replicase).

These results show that the spike protein (SP) and replicase proteins exhibit the most and highest quality similarities with human cardiac proteins among the SARS-CoV-2 components. The SP contains multiple sequences that mimic a range of cardiac proteins, including myosin, laminins, collagens, beta-2 adrenergic receptor (B2AR), angiotensin converting enzyme 2 (ACE2), and pyruvate kinase (Figure 1 and Figure 2). SARS-CoV-2 replicase also mimics myosin, collagens, B2AR, ACE2, and pyruvate kinase, as well as actin and creatine kinase (Figure 1 and Figure 3). Many of these similarities are of very high quality, sharing six or seven amino acids in a sequence of ten and exhibiting very high Waterman-Eggert (WE) scores (>60) and/or unusually low E values (significantly less than 1.0). These criteria are based on substantial research demonstrating that sequences exhibiting at least this degree of similarity have a >85% probability of being cross-reactive under experimental conditions [58,59,60,61]. The statistical significance of these values will be discussed below. The SARS-CoV-2 nucleoprotein and protein 3a account for most of the additional similarities observed to cardiac proteins, with the remaining SARS-CoV-2 proteins contributing a few high-quality matches.

Two types of information are required to interpret these results and the additional proteomic data that follow this section. First, it is necessary to know the probability that the quality of matches reported here may appear by chance. LALIGN contains several internal measures of such probability, including the Waterman-Eggert (WE) score and the E value. In general, WE scores above 50 and E values below 1.0 are considered statistically significant matches, as determined by the probability that these values will occur in a random search involving proteins of similar sizes against the entire protein database [62]. However, Waterman and Eggert explicitly caution users of LALIGN and BLAST to run their own study-appropriate controls as well [62].

A previous study using the same methodology employed here provided one set of study-appropriate controls. In the previous study, the likelihood of 13 polypeptides being similar to any of the 25 peptide receptors (325 comparisons) was calculated. Only 0.3% involved matches with an identity score of 7 out of 10, and only 1.8% achieved 6 out of 10. In contrast, 10.0% of the 216 SARS-CoV-2-cardiac protein comparisons exhibited a 7 out of 10 (or better) match, and 40.3% exhibited a 6 out of 10 match. Correspondingly, while none of the polypeptide-receptor pairings achieved a WE score above 50, 9 of 216 SARS-CoV-2-cardiac protein comparisons (4.1%) achieved WE scores of 70 or above, and 32 of 216 comparisons (14.8%) achieved WE scores between 60 and 70. Formal statistics are not required to observe that the differences between the polypeptide/receptor results and the SARS-CoV-2/cardiac protein results are significantly different. Notably, the majority of the high-quality matches involving SARS-CoV-2 were concentrated in the SP and replicase proteins (Figure 1).

An additional set of internal study controls is provided in Section 2.2.

### 2.2. Proteomic Study of Similarities Shared by Proteins of Other Viruses and Human Cardiac Proteins

LALIGN was also used to explore the number and quality of similarities between viruses other than SARS-CoV-2 and human cardiac proteins. The viruses chosen as controls included two that have previously been associated with AM and DCM as possible triggers of the autoimmunity–coxsackievirus type B3 (CXB3) [63,64,65,66] and adenovirus type 5 (Ad5) [66,67]–and three that have a questionable association with AM and DCM–hepatitis C virus (HCV) and poliovirus (PV)–or no known association: influenza A H1N1 [63,66,68]. The results, summarized in Figure 4, revealed that SARS-CoV-2 has significantly more similarities to heart proteins (Table 2) of higher quality (Table 3) than do any of the other viruses. Table 2 and Table 3 also show that the influenza A virus has significantly fewer high-quality matches with cardiac proteins compared with all of the other viruses. CXB3, HCV, and Ad5 displayed very similar numbers and qualities of matches to cardiac proteins, squarely between SARS-CoV-2 and influenza viruses. Because adenovirus vectors have been used to deliver some SARS-CoV-2 vaccines, and these vaccines have been found to have some risk of inducing cardiopathies (see Introduction), Figure 5 illustrates some of the best quality matches between Ad5 proteins and cardiac proteins.

### 2.3. Proteomic Study of Similarities Shared by Bacterial and Human Cardiac Proteins

Since severe COVID-19 patients are at higher risk for AD than mild cases, and since severe cases are much more likely to contract bacterial co- or super-infections (see Introduction), an analysis of possible similarities between the most common COVID-19 bacterial co- and super-infections and human cardiac proteins was performed. The LALIGN approach of comparing each cardiac protein to each virus protein was not possible since bacteria have thousands of proteins. Instead, each cardiac protein was compared to the total bacterial species’ proteomes using BLASTP. The same criteria utilized above were used here to evaluate significance: Only matches that had a Waterman-Eggert score of at least 50, an E value of less than 1.0, and contained a sequence of ten amino acids in which at least six were identical were counted as sufficiently similar to induce possible cross-reactive immunity; this criterion is based on substantial research demonstrating that sequences exhibiting at least this degree of similarity have a >85% probability of being cross-reactive under experimental conditions [58,59,60,61].

Figure 4 summarizes the results, while Figure 6, Figure 7, Figure 8 and Figure 9 provide examples of the matches found. These matches were often of better quality than the virus matches, both in terms of the number of identical amino acids in any given sequence and in terms of the length of the sequences.

Overall, the bacteria were more likely than the viruses to mimic myosin, actin, collagens, β2GPI, and the enzymes alpha enolase, creatine kinase, and pyruvate kinase, and the similarities tended to be more extensive than those displayed by the viruses (Figure 4, Figure 6, Figure 7, Figure 8 and Figure 9). The mimicry between the Streptococcal M protein (omitted in Figure 6, which emphasizes additional potential myosin mimics) has been extensively studied previously and employed as the basis for numerous animal models of rheumatic heart disease and autoimmune myocarditis [69,70,71,72]. However, it is notable that other bacteria, including *Enterococcus faecium, Staphylococcus aureus,* and *Klebsiella pneumoniae,* also displayed many significant similarities to cardiac myosin (Figure 4) that might also play roles in triggering autoimmune cardiopathies. The clinical picture is, however, somewhat confused by the fact that cytotoxic antibodies targeting the Streptococcal M protein and cross-reacting with cardiac myosin also cross-react with enteroviruses such as polio and coxsackieviruses [73,74,75]. In light of the data presented in Figure 4 that coxsackieviruses, polioviruses, and *Streptococci* each have proteins that mimic cardiac myosin, this cross-reactivity is perhaps not surprising.

In sum, the similarity searches yielded the interesting result that both viruses (especially SARS-CoV-2 and adenovirus type 5) and bacteria associated with severe COVID-19 displayed large numbers of high-quality and often extensive sequences of similarity with human cardiac proteins. Thus, both viral and bacterial antigens might play roles in triggering autoimmunity in COVID-19. The fact that coxsackieviruses, polioviruses, and HCV also display many cardiac protein similarities and are associated with some risk of autoimmune myocarditis is also consistent with these proteomic results.

### 2.4. Experimental Results of Virus Antibody Binding to Cardiac Proteins Using ELISA

To test the utility of the similarity results, polyclonal antibodies against viruses and bacteria associated with COVID-19 (e.g., SARS-CoV-2, adenoviruses, *Streptococci*, *Staphylococci*, *Enterococci*), and a random selection of microbes not associated with COVID-19 (e.g., influenza, coxsackieviruses, herpes simplex type 1, *Clostridia*, and *Mycobacterium tuberculosis*) were tested for their binding to cardiac proteins using quantitative ELISA.

The results of the virus antibody experiments are summarized in Table 4, and examples of the binding curves obtained are shown in Figure 10, Figure 11 and Figure 12. Most of the observed binding involved the SARS-CoV-2 spike protein or nucleoprotein. Unfortunately, no polyclonal antibodies were available against the replicase, which also displayed a large number of high-quality similarities to cardiac proteins in the similarity searches above. Among the control viruses, several induced antibodies that recognized myosin and adenovirus antibodies also recognized laminin, fibronectin, and creatine kinase. Coxsackievirus antibodies were notable for binding to myosin and laminin. Also notable is the result that no virus antibody tested bound to cardiolipin, although anti-cardiolipin antibodies are one of the most common found in COVID-19 patients with autoimmune cardiopathies (see Introduction).

The first set of experiments that we performed determined whether it was possible to substitute non-human proteins for human proteins in these studies, and, with one exception (actin, which is noted in the results that follow), the SARS-CoV-2 antibodies employed here bound (or did not bind) equally to both human- and animal-derived proteins. Thus, most of the studies were carried out with non-human proteins because they were available in greater quantities at lower prices.

Note that in order to interpret the significance of the binding constants derived from the curves illustrated in Figure 11, Figure 12 and Figure 13 (and in the Figures that follow in the next section), it is necessary to compare the binding constants to the concentration of cardiac protein present either in the heart or in blood serum. The relevant data are summarized in Table 5. As a general rule, the binding constant should be equal to or smaller than the concentration of the protein for the antibody to bind sufficient protein to impair its function or induce complement activation to cause cellular damage. Binding constants in Figure 10 that satisfy these criteria are bolded on a gray background, and the same formalism is used in the next section as well.

### 2.5. Experimental Results of Bacterial Antibody Binding to Cardiac Proteins Using ELISA

Bacterial antibodies against Staphylococcus aureus, Streptococci, Klebsiella pneumonia, Escherichia coli, Clostridia, Mycobacterium tuberculosis, and Enterococcus faecium, all of which are common co-infections in severe COVID-19 (see Introduction), were also tested for binding to cardiac proteins using the same quantitative ELISA protocol used to test the virus antibodies. The results are summarized in Table 6, and some of the binding curves are illustrated in Figure 10(Right), Figure 11(Left) and Figure 12(Left) above and in Figure 13 and Figure 14 below. Streptococci, Staphylococci, Klebsiella, and Enterococci antibodies were notable for high affinity binding to cardiolipin and pyruvate kinase, to which none of the SARS-CoV-2 or other virus antibodies bound significantly, as well as to myosin. None of the bacterial antibodies bound with significant affinity to B2AR or creatine kinase, which were targets of some of the SARS-CoV-2 and adenovirus antibodies (Figure 10). Since all of these proteins are targets of autoimmunity in some forms of COVID-19 autoimmune cardiopathies, these differences in virus and bacteria cross-reactivities may be important for understanding the microbial triggers involved in eliciting autoantibodies against different cardiac proteins.

### 2.6. Experimental Results of Virus Antibodies Binding to Bacterial Antibodies Using DA-ELISA

One of the unusual aspects of the previous results is that many of the targets of the virus and bacterial antibodies are cardiac proteins that are molecularly complementary to each other. For example, actin and myosin bind to form actinomyosin; the basement membrane of cardiomyocytes is formed by complexes of laminins and collagens; etc. The complementarity of the protein targets suggested that some of the antibodies cross-reacting with these proteins might also be complementary to each other, therefore acting like idiotype-anti-idiotype pairs. Double-antibody enzyme-linked immunoadsorption assays (DA-ELISAs) were used to explore this hypothesis, and a summary of the results is shown in Table 7.

Figure 15 illustrates some of the resulting binding curves. These curves are generally not as well formed as those resulting from antibody binding to pure antigens (previous Figures) because these experiments universally employed polyclonal antibodies, in which a range of possible antibody-antibody specificities could theoretically exist. Thus, rather than a nice “S”-shaped curve, the curves resulting from polyclonal-antibody/polyclonal-antibody binding are often, as in Figure 15, slowly rising curves or display multiple levels of binding. Binding constants for such curves (Table 6) are therefore necessarily general estimates derived from the mid-points of these complex curves and are provided as general guides to the antibody-antibody affinities rather than as firm measures of specific affinity.

The results of the DA-ELISA experiments show that *Enterococcus, Streptococcus, Staphylococcus,* and *Klebsiella* bind with significant affinity to antibodies against spike protein antigens of SARS-CoV-2 but not to other SARS-CoV-2 proteins. This result is consistent with the spike protein, and these particular bacteria each display the majority of cardiac protein similarities. Notably, adenovirus antibodies did not bind to any of the bacterial antibodies, nor did influenza virus antibodies. Coxsackievirus antibodies, however, displayed an affinity for many of the bacterial antibodies, including several that the SARS-CoV-2 antibodies did not. Thus, viruses that are highly associated with autoimmune cardiopathies (SARS-CoV-2 and coxsackieviruses) produce antibodies that bind to bacteria associated with an unusually high risk of such cardiopathies (*particularly Streptococci*), while viruses unrelated to autoimmune cardiopathies seem to lack this property.

## 3. Discussion

### 3.1. Summary

The results of this study demonstrate that SARS-CoV-2 proteins mimic human cardiac proteins to a significantly greater degree than do other viruses such as Ad5, coxsackieviruses, or HCV that are associated with the risk of autoimmune cardiopathies, or influenza A H1N1 virus, which is not. Some antibodies against SARS-CoV-2 proteins, particularly those against elements of the spike protein, cross-react with some of these cardiac proteins, particularly myosin, actin, collagens, laminins, and the beta 2 adrenergic receptor. Adenovirus antibodies also recognized some cardiac proteins, including myosin, laminin, fibronectin, and beta 2 glycoprotein I. Additionally, coxsackievirus antibodies cross-reacted with myosin, actin, and collagen I, as previously reported [59,91]. The statistically significant increase in cardiac protein matches displayed by SARS-CoV-2 and Ad5 over PV, InfA, and HCV (Table 2) is similar to a previous study concerning similarities between these viruses and coagulopathy-related proteins [61,92]. Thus, SARS-CoV-2 increases the risk not only of autoimmune cardiopathies but also of autoimmune coagulopathies that can lead to cardiac arrest and stroke. The data concerning adenovirus is also of particular interest since Ad5 is used as a delivery vector for some SARS-CoV-2 spike protein vaccines under development (e.g., CanSino Biologics Inc. and Beijing Institute of Biotechnology, Ad5-nCOV; ImmunityBio, Inc. and NantKwest Inc., hAd5-S-Fusion+N-ETSD; Vaxart, VXA-COV2-1; Altimmune, Inc., AdCOVID), while other variants of adenoviruses are employed in the Jannsen/Johnson and Johnson (Ad26.COV2-S; Human, Ad26) and the Oxford-AstraZeneca (ChAdOX1-nCoV; Chimpanzee, ChAdY25) [93]. It is likely that adenovirus vectors increase the probability of post-vaccinal autoimmune sequelae, and other virus vectors with lower risks, as determined by a combination of similarity and cross-reactivity studies, might be worthy of study.

The bacteria studied here, which were chosen for their frequency as co-infections with SARS-CoV-2 in hospitalized COVID-19 cases, also displayed a large number of similarities with cardiac proteins, many of which were confirmed by antibody cross-reactivity. Notably, the number of SARS-CoV-2 similarities to cardiac proteins pales in comparison with *Klebsiella pneumoniae* and is no greater than *Streptococcus pneumoniae* or *Enterococcus faecium* (Figure 1), suggesting that any of these bacteria could have at least as high a probability of inducing an autoimmune cardiopathy as the virus. However, the proportion that these similar antigens comprise within SARS-CoV-2 is far greater than among any bacterium because SARS-CoV-2 is comprised of only thirteen proteins (and many COVID-19 vaccines, only the SARS-CoV-2 spike protein), while the average bacterium has thousands. Thus, the probability that the host immune system will preferentially respond to a SARS-CoV-2 antigen that mimics a heart protein is presumably much greater than the probability that it will respond to any particular bacterial antigen. The quality of the mimic, meaning the degree to which the protein mimics share some or all of their amino acid sequences, must also be a determinant of whether any particular mimic has the potential to induce autoimmunity. In this context, the very large number of very high-quality matches (an E value greater than or equal to 60) among SARS-CoV-2 and Streptococcal antigens as compared with the relatively few high-quality matches displayed by *E. faecium* may be an important factor to consider as determinants of autoimmune disease risk (Figure 2).

### 3.2. Comparison with Previous Study Results

The proteomic results presented here are similar to those found by other groups, although this study has used a different set of criteria for performing the searches and evaluating their significance. Previous studies have also identified a range of cardiac proteins, including tropomyosin and actin-binding proteins [94], ACE-2 and ANCA [95], and myosin [96,97], as potential targets of SARS-CoV-2 antibodies but not actin itself, collagens, laminins, adrenergic receptors, alpha enolase, or creatine or pyruvate kinases. Additionally, this is the first study to investigate whether bacterial infections associated with SARS-CoV-2 might also play a significant role in triggering cardiac autoimmunity in COVID-19.

As can be seen in Table 8, the existence of high-quality similarities, or their total number, between a microbe and cardiac proteins does not guarantee antibody cross-reactivity. Overall, the likelihood of inducing autoimmunity is presumably a function of at least three factors: the number of similar sequences a microbe displays, the proportion of antigens these similar sequences comprise, and the quality of the similarities. However, viruses with many high-quality matches do tend to induce polyclonal antibody responses that cross-react with the proteins to which they are most similar. In terms of sheer numbers, SARS-CoV-2 displays the most similarities to heart proteins of any virus tested and the most cross-reactivity with heart proteins exhibited by its antibodies. The same correlation is not as well observed among the bacteria, perhaps because, as noted above, the bacteria present a much larger number of antigens to the immune system, which must then target the most antigenic. It is also notable that while the sets of cardiac antigens that are recognized by the microbial antibodies overlap to some extent among the viruses and bacteria (e.g., myosin and collagens), the bacteria elicit a number of antibodies that cross-react with cardiac proteins that the viruses do not; these include cardiolipin, pyruvate kinase, fibronectin, and beta 2 glycoprotein I. Thus, the range of cardiac antigens targeted in any particular patient may depend on the virus, bacterium, or their combination.

Similarities were found in the proteomic searches (Figure 1) and in the targets of autoimmunity observed in human patients (summarized in Table 9) that could not be tested for cross-reactivity here due to the limited availability of sufficient protein at affordable costs. These additional protein targets, which have been documented in both pre-COVID autoimmune cardiomyopathies and in COVID-19 autoimmune cardiopathies, included angiotensin converting enzyme 2 (ACE2) [51,98,99,100], antineutrophil cytoplasmic antibodies (ANCA) [101,102,103], alpha enolase [104], alpha- and beta-adrenergic receptors [42,105,106,107,108,109], laminins [110], phospholipids [111,112,113], tropomyosin [114], and troponin [104,115,116]. Table 9 summarizes the known autoantigens that are related to these cardiopathies in severe and hospitalized COVID-19 patients. Table 9 also summarizes and compares these clinical findings to our results and those of Vojdani et al.’s [117,118] testing of SARS-CoV-2-induced rabbit polyclonal antibodies and human monoclonal antibodies against cardiac-related proteins. There is a good correspondence between the human and rabbit results. This comparison demonstrates that antibody cross-reactivities fall into three groupings. One group involves cross-reactivities between SARS-CoV-2 antigens and human cardiac antigens that are not recognized by any of the anti-bacterial antibodies tested. The second group involves cross-reactivities to cardiac proteins by both SARS-CoV-2 and bacterial antibodies. And the third group consists of ant-bacterial antibodies against cardiac antigens that are not recognized by any of the SARS-CoV-2 antibodies tested (Table 9) [42,51,98,99,100,101,102,103,104,105,106,107,108,109,110,111,112,113,114,115,116,117,118,119,120,121,122,123,124,125]. The importance of these groupings is two-fold. First, it is evident that both SARS-CoV-2 and bacterial antigens can participate in inducing antibodies cross-reactive to cardiac autoantigens; and second, to account for the range of autoantigen targets observed in hospitalized COVID-19 patients, both SARS-CoV-2 and bacterial infections are required. For example, antibodies known to target actin, adrenergic receptors, phospholipids, and tropomyosin are induced by SARS-CoV-2 but not (as far as current studies indicate) by bacteria. On the other hand, no study has thus far reported that SARS-CoV-2 antibodies cross-react with β2GPI, cardiolipin, alpha enolase, fibronectin, or pyruvate kinase, while all five of these proteins are known to be targets of bacterial antibodies. (Note that although alpha enolase was not tested for cross-reactivity in this study, numerous very high-quality proteomic similarities to bacterial enolases are reported here, and previous experimental studies [119,120,121] have reported cross-reactivity between these bacterial enolases and human enolases.) Finally, there are a number of cardiac proteins targeted by microbial antibodies, including collagens, creatine kinase, laminins, mitochondrial antigens, and myosin, that may be induced by either SARS-CoV-2 and/or bacterial antigens.

### 3.3. Evidence of Antigenic Complementarity between SARS-CoV-2 and Bacteria

To summarize thus far, proteomic similarity studies suggest that SARS-CoV-2 as well as bacteria associated with increased severity of COVID-19 both display unusual numbers of high-quality similarities to human cardiac proteins that in vitro and clinical studies generally confirm. One additional factor also implicates a combination of virus and bacteria in the induction of autoimmune cardiopathies, and that factor involves the complementarity of the sets of SARS-CoV-2 and bacterial antibody targets described in Table 9. A majority of these protein targets are complementary to at least one of the others so that they bind to each other as part of their normal physiological function (Figure 16). Actin and myosin bind to each other to form actinomyosin [126], and these proteins can further bind troponin [127,128]. Troponin can also bind to laminin [129], as can alpha-enolase [130]. Indeed, an alternative name for bacterial enolases is “laminin binding protein” (e.g., Figure 8: UniProtKB, O69174.1) [131,132,133]. Alpha enolase can additionally bind strongly to fibronectin, fibrinogen, and collagen type IV [130] and is expressed on the cell surfaces of cardiomyocytes [134]. Laminins and collagens also bind to each other to form the extracellular matrix [135,136]. Connectin binds to both laminin and actin [137]. Pyruvate kinase binds to extracellular matrix proteins, including laminins and fibronectins [138]. And finally, creatine kinase and pyruvate kinase bind to each other [139,140,141] (Figure 17). Because of these overlapping affinities, some of these proteins are characterized by conserved binding regions, so that antibodies against myosin also recognize collagens [59,91,142]; antibodies against actin also recognize laminins [59,91], and antibodies against troponin also recognize alpha enolase [104] (Figure 16).

These binding complementarities are significant for two reasons in the context of autoimmune cardiopathies. First, just as SARS-CoV-2 utilizes the complementarity between its spike protein and ACE-2 to infect specific cell types, so the bacteria associated with COVID-19 utilize the complementarity between their extracellular protein antigens, such as enolase and pyruvate kinase, and extracellular matrix proteins, such as laminins, collagens, and fibronectins [143,144], on human tissues. Thus, some of the main targets of autoimmune cardiopathies are proteins such as ACE2 that mediate SARS-CoV-2 infection, while other targets are proteins mediating bacterial infections. Secondly, SARS-CoV-2 and adenoviruses mimic some cardiac proteins, while COVID-19-associated bacteria mimic complementary sets of proteins. Thus, combinations of viral and bacterial infections are highly likely to elicit antibodies that are complementary to each other, just as the inducing proteins are complementary to each other. This complementarity of microbial antigens and their human protein mimics likely explains the complementarity between SARS-CoV-2 antibodies and those from some of the bacteria that were observed in the DA-ELISA experiments above (Figure 17, Figure 18 and Figure 19). The resulting idiotype-anti-idiotype relationship is actually the result of a pair of complementary idiotypic responses and may explain the observation that circulating immune complexes (CIC) are very frequent concomitants of autoimmune cardiopathies in general [145,146,147,148], including approximately 80% of severe cases of COVID-19 [149,150], the multisystem inflammatory syndrome in children (MIS-C) [151] and COVID-19 vaccine-related cardiopathies [152].

### 3.4. Proposed Model of Autoimmune Cardiopathy Induction

In short, the data aggregated here suggests that COVID-19 cardiopathies are induced in the following manner. SARS-CoV-2 and bacterial antigens cooperate to induce sets of antibodies that target complementary cardiac antigens and that act like idiotype-anti-idiotype pairs. Among the complementary sets of antigens known to bind to each other are actin and myosin, troponin and myosin, collagen and laminin, alpha enolase and laminin, pyruvate kinase, and collagen, etc. The virus utilizes ACE2 to infect tissues, while the bacteria utilize extracellular matrix proteins such as collagens, laminins, and fibronectin. A simplified cartoon of some of these relationships is provided in Figure 17.

**Figure 17 ijms-24-12177-f017:**
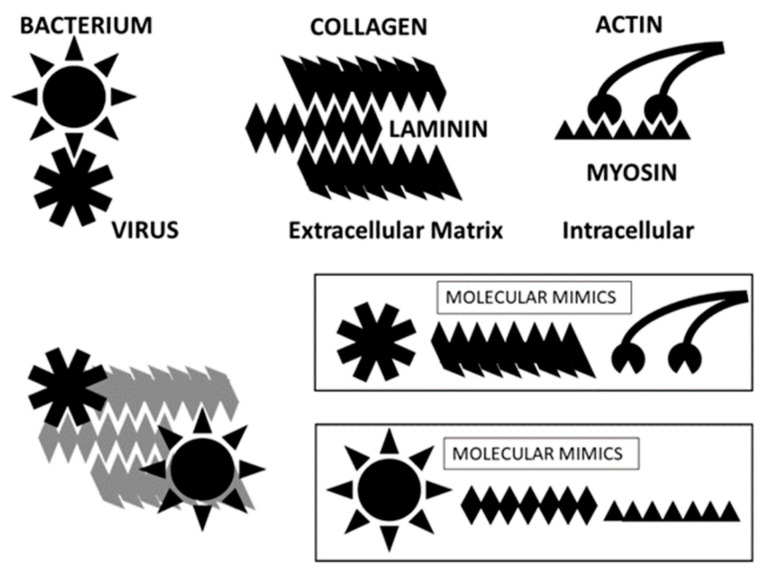
Cartoon illustrating some key sets of complementary antigens and molecular mimics presented in COVID-19 autoimmune cardiopathies. SARS-CoV-2 and adenoviruses (“virus”) induce antibodies that are complementary to (in the sense of idiotype-anti-idiotype) antibodies against several bacteria that are common co- or super-infections of hospitalized COVID-19 patients, including Streptococci, Staphylococci, Klebsiella, and Enterococci (“Bacterium”). Collagen and laminin bind to each other to form the basement membrane of cardiomyocytes. Bacteria and viruses utilize laminin or collagen to bind to and infect cells. Actin and myosin bind to each other to form the active contractile element of muscles, actinomyosin. Data presented here and summarized from previous studies demonstrate that SARS-CoV-2 induces antibodies that also recognize collagen and actin. Bacteria induce antibodies that cross-react with lamin and myosin. Thus, significant complementarity and mimicry exist within the sets of antigens associated with cardiac complications present in hospitalized COVID-19 patients. The set of complementary antigens and antigen mimics illustrated here is not intended to be complete, and many additional examples are provided above, especially in Table 9.

The immune system responds to viral and bacterial infections by inducing two sets of antibodies, one against each type of microbe. Antibodies against the bacterium will cross-react with cardiac proteins such as laminins, fibronectin, and alpha enolase that mimic bacterial proteins. Antibodies against the virus will cross-react with cardiac proteins such as collagens, adrenergic receptors, or creatine kinases that mimic virus proteins. Because the autoantigens targeted by the virus and bacterial antibodies are themselves complementary and bind to each other, some of the antibodies against the virus and bacterium will also be complementary to each other, behaving like idiotype-anti-idiotype pairs to form CIC (Figure 18).

**Figure 18 ijms-24-12177-f018:**
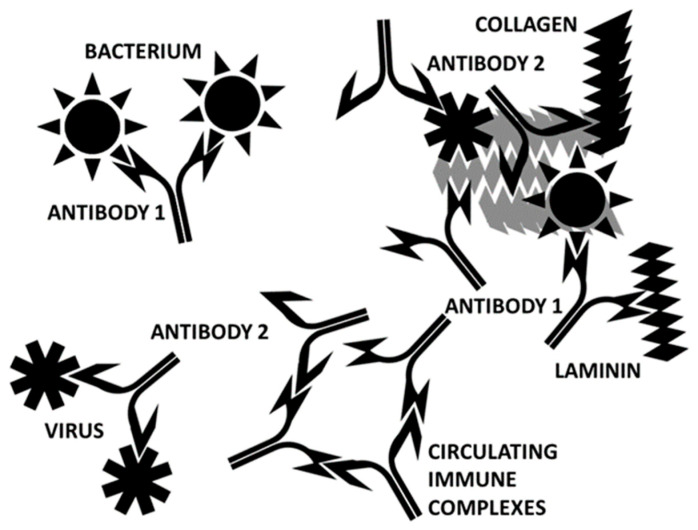
Cartoon illustrating the initiation of autoimmune cardiopathies. A virus that expresses antigens that mimic human cardiac proteins (SARS-CoV-2 and adenoviruses, in this case) elicits a set of antibodies (“antibody 2”). A bacterial co- or super-infection that mimics complementary cardiac proteins induces another set of antibodies (“antibody 1”). Some of these antibodies are complementary to each other (in the sense of idiotype-anti-idiotype) and bind to each other to form circulating immune complexes. The binding of viruses and bacteria to complementary antigens on cardiomyocytes results in the release of cardiac proteins, some of which are recognized by the microbe-induced antibodies because of molecular mimicry. The immune system is now unable to distinguish “self” from “non-self” because each microbe-induced antibody not only cross-reacts with a self-protein because of mimicry but also recognizes the other set of antibodies as targets as well.

Antibody binding to cell-surface antigens on cardiomyocytes, such as laminins, collagen, ACE2, adrenergic receptors, etc., as well as the presence of CIC, will initiate complement activation, resulting in destabilizing cardiomyocyte integrity. In consequence, cardiomyocytes will release previously hidden antigens such as actin, myosin, cardiolipin, and other mitochondrial antigens. These additional autoantigens also mimic some of the viral and bacterial antigens initiating the autoimmune response, provoking additional autoimmune activity and increased CIC production (Figure 19). It is likely that there is epitope drift in the antibody specificities due to the higher antigenicity of these hidden antigens.

**Figure 19 ijms-24-12177-f019:**
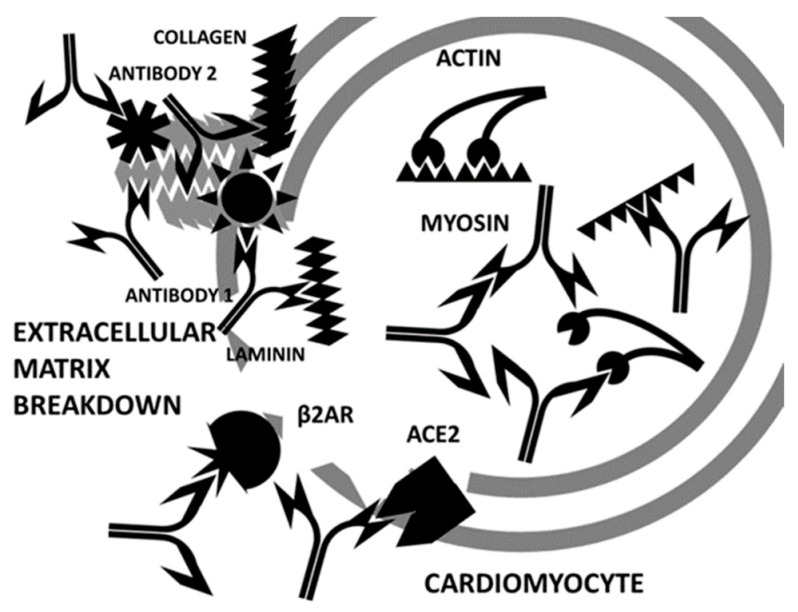
Cartoon illustrating the second stage of the autoimmune attack on cardiomyocytes. Disruption of cardiomyocyte integrity by the attack on cross-reactive antigens such as collagen, laminin, adrenergic receptors (“b2AR”), or angiotensin converting enzyme 2 (“ACE2”) results in the release of so-called “hidden antigens” such as myosin and actin that become major targets of autoimmune disease due to epitope drift. The complementarity of these “self” antigens continues to drive the production of complementary antibodies, furthering the confusion between “self” and “non-self”.

Finally, it is important to realize that the antigen complementarity driving the antibody complementarity results in each antibody mimicking one of the antigens (Figure 20). Thus, the antibody against the bacterium may mimic a virus antigen, and the antibody against the virus may mimic a bacterial antigen. Additionally, each antibody will therefore mimic some of the host autoantigens as well. The overall result is to confuse the distinction between “self” and “non-self”. Since the specific heart proteins that each microbe mimics differ from one bacterium to another, these differences are likely to have consequences for the specific targets that are attacked should autoimmunity be induced, and thus the range of different symptoms that confuse the COVID-19 picture.

Additional consequences of targeting complementary antigens create an ongoing autoimmune disease process. The co-processing of complementary antigen complexes, such as laminin with collagen or myosin with actin, results in altered proteasome products that have higher autoimmunogenicity than occur when the same antigens are processed singly (reviewed in [153]). Virus antigens and bacterial antigens, and their host damage-associated molecular pattern (DAMP) mimics, each activate sets of toll-like receptors (TLR) and nucleotide-oligomer-domain-like receptors (NLR) that synergize to produce the hyperinflammatory conditions that support or drive the autoimmunity; CIC can also drive innate activation (reviewed in [154,155]). Additionally, not only are complementary (idiotype-anti-idiotype) antibodies produced in response to complementary virus-bacterium infection, but complementary T cell receptors (TCR) also appear to be induced in COVID-19 autoimmune complications such as cardiopathies, coagulopathies, and MIS-C [156]. Moreover, these TCR pairs, because they mimic complementary antigens, each mimic one of the antigens, so that the immune system becomes complicit in driving ongoing autoimmunity [156]. Thus, induction of complementary sets of antibodies (and TCR) by complementary sets of antigens that cross-react with complementary sets of autoantigens creates all of the conditions within the innate and adaptive immune systems necessary to overcome “self” tolerance and to support an ongoing autoimmune disease process [52,53,157,158,159,160].

The model is generalizable. We have previously demonstrated that SARS-CoV-2 can interact with some of the bacteria studied here to provoke autoimmune coagulopathies directed against blood proteins [61,92] and a similar approach might illuminate the causes of other autoimmune complications associated with SARS-CoV-2 infections. The basic hypothesis is that different strains of SARS-CoV-2 combined with different bacterial co- or super-infections are likely to trigger different types of autoimmunity depending on the particular sets of complementary antigens the microbes display in common with their host. It is additionally likely that the risk of any particular type of autoimmune complication is partially determined by host genetic variations, as has been found for all other autoimmune diseases. Additionally, a very similar model has independently been proposed to explain the etiology of anti-neutrophil cytoplasm antibody (ANCA) vasculitis as a result of the immune response to proteinase 3, collagen, and their complementary antigens, which include plasminogen [161,162,163,164,165,166]. Thus, vascular autoimmunity following COVID-19 may be induced by a mechanism similar to the one described here.

The model describing the induction of autoimmune cardiopathies in COVID-19 applies equally to understanding the rare instances of such cardiopathies following SARS-CoV-2 vaccination, other COVID-19 autoimmune complications, and autoimmune cardiopathies that occur in conjunction with other virus and bacterial infections. Given the large number of high-quality similarities between the SARS-CoV-2 spike protein that is present in all current COVID-19 vaccines, it is evident that the simple presence of molecular mimicry is not sufficient to induce autoimmunity since the vast majority of infected and vaccinated people do not develop such complications. The extraordinary rarity of such complications argues instead for the necessity of some type of cofactor: for example, a complementary bacterial infection coinciding with the vaccination period. Since most people receiving vaccinations are healthy, the incidence of such complementary bacterial infections (as opposed to any microbial infection whatever) can be assumed to be extremely low. However, one implication of the present study is that it may be possible to decrease the risk even further by screening people who are to receive vaccines for fever and the presence of *Staphylococcal, Streptococcal, Klebsiella*, and *Enterococcus* bacterial infections, whether oropharyngeal, respiratory, nasal, gastrointestinal, urogenital or involving the skin. Additionally, vaccination against streptococcal infections by means of pneumococcal vaccines has been demonstrated in numerous studies to lower the risk of hospitalization and death during the COVID-19 pandemic [167,168,169,170,171,172,173,174,175,176] and may do so, at least in part, by decreasing the risk of autoimmune complications.

Our data are also important for revealing significant risks for autoimmune cardiopathies associated with adenoviruses and coxsackieviruses (Figure 19). The risks of autoimmune cardiopathies associated with both viruses have been documented previously (coxsackieviruses [52,53,59,91,177,178,179]; adenoviruses [177,178,179], but, as with SARS-CoV-2, the proportion of affected individuals is extremely small compared with the numbers of people infected. The model proposed above may, again, provide a possible explanation: these viruses may need the presence of appropriate bacterial co- or super-infections to transform their molecular mimicry of cardiac proteins into an unregulated attack on “self”. This proposition has previously been examined in the case of coxsackievirus-associated autoimmune myocarditis, which is highly associated in some studies with concurrent infections of coxsackieviruses and group A Streptococci (e.g., [180,181,182,183,184,185]; and reviewed in [59,91]. The possibility that other bacteria may synergize with other cardiotropic viruses such as adenoviruses has not, apparently, been examined. and can stand as a prediction made by the model presented here. The possible role of adenoviruses in promoting the increased risk of such autoimmune complications as either co-infections of SARS-CoV-2 or as vectors for SARS-CoV-2 vaccines is clearly warranted given the greater proportion of vaccinees who develop cardiopathies following adenovirus-vectored vaccines than those receiving nanoparticle-delivered vaccines [22,23,24].

### 3.5. Limitations

This study has a number of limitations. One is that it has used non-human antibodies that may or may not reflect the range of specificity of human polyclonal antibodies. On the other hand, Table 9 demonstrates that the results reported here do correlate well with the results of clinical studies and previous studies of cross-reactivity using human monoclonal antibodies [118]. A second limitation is the limited range of cardiac proteins that were tested for cross-reactivity. ACE2, alpha-enolase, and troponin are among the particularly important proteins that warrant further investigation as targets of autoimmunity observed in clinical studies, but the expense of these proteins created a barrier to the investigation here. Mitochondrial antigens were also ignored in this study but represent a target in autoimmune cardiopathies identified in both clinical and in vitro studies (Figure 20). A third limitation of this study is the lack of an animal model to test the hypothesis that autoimmune disease is induced by combinations of viral and bacterial antigens. Notably, however, two groups have previously demonstrated that coxsackieviruses combined with streptococcal antigens do induce autoimmune cardiopathies in rodents [181,186], and *Klebsiella* antigens have been successfully used in combination with heart proteins to induce autoimmune myocarditis in mice [187], which makes the prediction that SARS-CoV-2 antigens in combination with these bacteria will do the same.

### 3.6. Further Tests

Many of the implications of our data can be tested further in animal models. For example, SARS-CoV-2-susceptible species (such as golden hamsters or some strains of mice) might be co-infected with the virus and with bacteria such as group A Streptococci, Staphylococci, Klebsiella, or Enterococci, and in combination with adenoviruses. The effect of such bacterial coinfections on vaccination with SARS-CoV-2 vaccines could be tested similarly. Our prediction is that animals infected with only SARS-CoV-2 or its vaccine or with the bacterium alone will not develop cardiopathies, while coinfected animals will demonstrate increased rates of cardiopathies. Alternatively, since it is assumed here that COVID-19 coagulopathies are autoimmune diseases, it should be possible to inoculate naive rabbits with combinations of polyclonal (rabbit) antibodies against the SARS-CoV-2 proteins (e.g., spike, nucleoprotein, or whole virus) in combination with (rabbit) polyclonal antibodies against group A Streptococci, Staphylococci, Enterococci, etc. Such combinations are predicted to produce clinically evident cardiopathies. Correspondingly, we predict that rabbits inoculated with only the SARS-CoV-2 antibodies or only the bacterial antibodies will not develop cardiopathies.

Further clinical studies are also needed. Are people who have been recently vaccinated against pneumococci less likely than those who are unvaccinated or who have not been vaccinated since childhood to develop autoimmune cardiomyopathies following COVID-19? Do people who develop such autoimmune cardiomyopathies have evidence of one or more of the bacterial infections identified here as complementary inducers of autoimmunity? Are hospitalized COVID-19 patients who are treated with broad-spectrum antibiotics at admission less likely to develop autoimmune complications than those who are so treated later in their disease course or those who are not given antibiotics (see, e.g., [187])? Do broadly acting inhibitors such as melatonin (reviewed in [44,154,155]) and steroids [188,189] that work simultaneously against multiple innate receptors to moderate the hyperinflammation that accompanies COVID-19 also lower the risk of autoimmune complications? Much remains to be known.

## 4. Materials and Methods

### 4.1. Similarity Searches

Two types of similarity searches were carried out to identify likely molecular mimics shared by SARS-CoV-2 proteins (accessed on 2 May 2021 from https://viralzone.expasy.org/8996) and human myocardial proteins (sequences accessed 2 May 2021 from UniProtKB https://www.uniprot.org/help/uniprotkb). The first type of search utilized BLASTP (version 2.2.31+) on the www.expasy.org server. BLOSUM80 was used to identify the type of short, continuous sequences approximately ten to fifteen amino acids in length that are presented by human leukocyte antigens (HLA) to T and B cells [47,48]. The E value was set to 1; filter low-complexity regions on; no gaps; 3000 best scoring and best alignments to show. Only matches that had a Waterman-Eggert score of at least 50, an E value of less than 1.0, and contained a sequence of ten amino acids in which at least six were identical were counted as sufficiently similar to induce possible cross-reactive immunity; this criterion is based on substantial research demonstrating that sequences exhibiting at least this degree of similarity have a >85% probability of being cross-reactive under experimental conditions [58,59,60,61].

The second search method employed LALIGN (www.expasy.org, accessed on 17 February 2022–14 May 2023) to do a deeper dive into the SARS-CoV-2 protein (accessed on 2 May 2021 from https://viralzone.expasy.org/8996) and identify similarities identified by the BLAST searches. The search algorithm was set to BLOSUM80, with a gap penalty of −10.0 and an E value of 10. The 20 best matches were displayed. The control viruses were poliovirus type 1, coxsackievirus B3, hepatitis A virus, rhinovirus 2, adenovirus 5, and influenza virus H1N1 (Wilson). UniProt accession numbers for the viruses and for the human myocardial proteins, as well as a list of the blood proteins, are provided in the table captions. As with the BLAST searches and for the same reasons, the LALIGN results were further culled for sequences with E < 1, Waterman-Eggert score > 45, and sequence similarity having a region containing at least six out of ten identities. The number of matches simultaneously satisfying the E value, Waterman-Eggert, and 6-of-10 criteria was tabulated, and representative matches were provided. We note that LALIGN on the Expasy server was decommissioned after we had completed this study. We find that running protein BLAST on the NCBI server https://blast.ncbi.nlm.nih.gov/Blast.cgi?PROGRAM=blastp&PAGE_TYPE=BlastSearch&LINK_LOC=blasthome (accessed on 7 July 2023) with the following settings yields very similar results but without providing Waterman-Eggert scores: BLOSUM80. E = 100, word size 3, existence 8; extension −2; eliminate low-complexity regions.

Cardiolipin could not be searched using either BLAST or LALIGN since it is not a protein, but its presence in each bacterium was determined from existing experimental literature [190,191].

### 4.2. Statistics

Statistics were applied to the tabulated LALIGN results using a paired T-test to explore pairwise comparisons between each class of virus-human protein combination and every other (https://www.graphpad.com/quickcalcs/ttest2/ (accessed on 7 July 2023)). Since all possible permutations of the results were explored, a Bonferroni correction was applied to the resulting *p* values (https://www.easycalculation.com/statistics/bonferroni-correction-calculator.php (accessed on 7 July 2023)). To satisfy *p* = 0.05 after a Bonferroni correction for the 10 pairwise comparisons made between the various viruses, the uncorrected *p* value had to be <0.005 (T > 3.0) to satisfy *p* = 0.05.

### 4.3. Experimental Protocols

ELISA and double-antibody ELISA (DA-ELISA) were employed to investigate whether the similarity searches yielded immunologically valuable information.

An enzyme-linked immunosorbent assay (ELISA) was used to investigate cross-reactivities between microbial antibodies and cardiac tissue-related proteins. The tissue protein was diluted in pH 7.4 phosphate buffer to a concentration of 10 µM. This standard solution was then diluted by ten-fold steps to about 10–14 M. Two wells received only phosphate buffer as controls. 100 µL of each protein dilution was added in duplicate to the wells of a Costar round-bottomed 96-well ELISA plate and incubated for one hour. The excess protein was triple washed out using a 1% Tween 20 solution (in phosphate buffer) and a plate washer. Next, 200 uL of blocking agent (2% polyvinylalcohol in phosphate buffer) was added to every well, incubated for an hour, and then triply washed. An antibody against a microbe (at 1 mg/mL concentration) was then diluted to 1/200 in phosphate buffer and 100 uL added to every well. The antibody was incubated for an hour and then triply washed. A species-appropriate horseradish peroxidase-linked secondary antibody was then diluted to 1/1000, incubated for an hour, and triply washed. Finally, 100 uL of ABTS reagent (Chemicon) was added, and incubated for 30 min, and the plate read at 405 nm in a Spectramax UV-VIS scanning spectrophotometer. The data were gathered using Spectramax software (version 4.0) and then analyzed using Excel. Analysis essentially consisted of subtracting non-specific binding to the buffer-only wells from the protein-containing wells and plotting the amount of antibody binding (as measured by absorbance at 405 nm) as a function of protein concentration.

Double antibody ELISA (DA-ELISA) was used to investigate possible antigenic complementarity between the antibodies used in the study. DA-ELISA differs from ELIAS in that the protein laid down in n the 96-well plate in the initial step of an ELISA is substituted with an antibody. A second antibody (from a different species) is tested for its ability to bind to the first. The ability of the second antibody to bind to the first is then monitored using peroxidase-linked antibodies against the species from which the second antibody is derived [90,91] As in the ELISA protocol, the first antibody is made up at a concentration of about 10 µM (assuming IgG antibodies have a molecular weight of 180,000 daltons) and then serially diluted by factors of ten. The rest of the protocol is the same.

### 4.4. Antigens

A list of the antigens utilized in experiments is provided in Table 10. The choice of antigens was determined by two factors: (1) whether the antigen is a known target of autoantibodies in COVID-19 cardiomyopathies (see Introduction, Table 1), and (2) the cost of the antigen. Given the large number of quantitative ELISA binding studies performed, some proteins known to be targets of autoantibodies in COVID-19 cardiomyopathies were too expensive for our budget (e.g., angiotensin II receptor; tropomyosin). Where possible and economically feasible, human proteins were utilized.

### 4.5. Antibodies

A list of the antibodies used in experiments is provided in Table 11. The choice of antibodies was determined by several factors: (1) polyclonality (autoimmune diseases result from polyclonal antibody activation, not monoclonal specificity); (2) availability (e.g., polyclonal antibodies against SARS-CoV-2 replicase protein could not be located); and (3) (especially for use in DA-ELISA experiments where different host species are needed unless one is an HRP-conjugated version—see Section 4.3 above), the host species.

## 5. Conclusions

The evidence compiled here strongly suggests that autoimmune cardiopathies following COVID-19 occur mainly in severe cases associated with bacterial co- or super-infections. SARS-CoV-2 itself mimics some of the known cardiac proteins targeted in such autoimmune cardiopathies, while bacteria such as Streptococci, Staphylococci, Klebsiella, and Enterococci mimic an overlapping but largely complementary set of cardiac proteins. Combinations of SARS-CoV-2 with various bacteria therefore result in the production of complementary sets of antibodies that not only cross-react with sets of complementary proteins such as collagens, laminins, and fibronectins, or myosins, actins, and troponins, but also with each other to form idiotype-anti-idiotype immune complexes. The result of this complex set of mimics and complements is the activation of complementary immune responses that lose the ability to distinguish “self” from “non-self”. The complementarity of the antigens, as well as the production of immune complexes, further drives an ongoing hyperactivation of the innate immune system.

The implications of these findings have practical value. One is that vaccination against bacterial co- or super-infections in COVID-19 should substantially decrease the risk of autoimmune cardiopathies, as should the timely and appropriate use of antibiotics for high-risk individuals. A second is that the risk of post-vaccinal cardiopathies may similarly be decreased in people recently vaccinated against pneumococci and screened for potentially complementary bacterial infections at the time of vaccination. A third is that it may be possible to implement new animal models of autoimmune cardiopathies employing combinations of viruses such as SARS-CoV-2, adenoviruses, or coxsackieviruses in combination with appropriate complementary bacteria such as Streptococci, Staphylococci, or Enterococci. Such models might permit novel treatments to be developed.

## Figures and Tables

**Figure 1 ijms-24-12177-f001:**
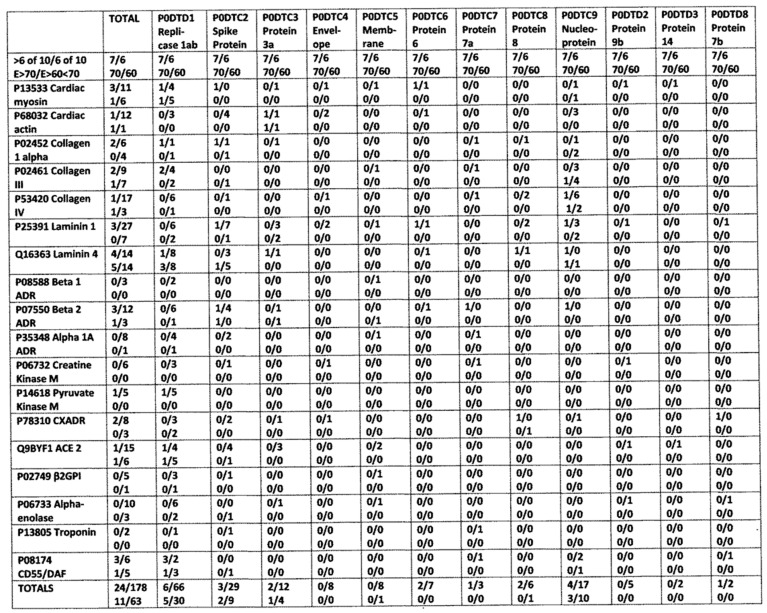
Summary of the number of LALIGN matches of SARS-CoV-2 proteins with cardiac proteins satisfying various criteria: TOP NUMBER LEFT in each box is the number of matches with that protein in which there were seven or more identical amino acids in a sequence of ten; TOP NUMBER RIGHT in each box is the number of matches with that protein in which there were six identical amino acids in a sequence of ten; BOTTOM NUMBER LEFT in each box is the number of matches that had an E score of 70 or greater; BOTTOM NUMBER RIGHT in each box is the number of matches that had an E score of 60 or more but less than 70. A number of matches meeting each criterion is summed for each cardiac protein (second column from the left) and for each SARS-CoV-2 protein (bottom row).

**Figure 2 ijms-24-12177-f002:**
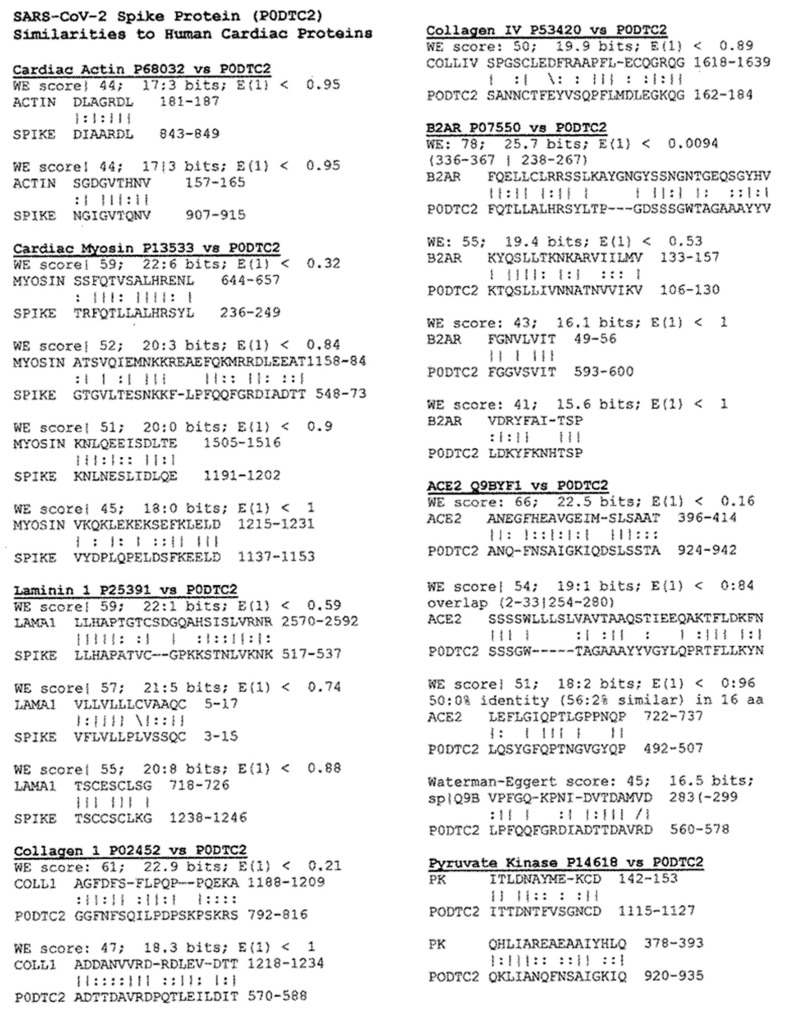
Examples of LALIGN-derived SARS-CoV-2 spike protein (P0DTC2) similarities to human cardiac proteins. Vertical lines indicate identities between the amino acids; double dots indicate conserved amino acid substitutions. WE stands for Waterman-Eggert score (the larger the score, the less likely the similarity is due to chance). E is a measure of probability inverse to the WE score, so the smaller E, the less likely the match is due to chance. The numbers following the protein name are the UniProtKB identifiers. Numbers following the amino acid sequence (or in parentheses before the sequence) define the place of the match in the amino acid sequences of the protein.

**Figure 3 ijms-24-12177-f003:**
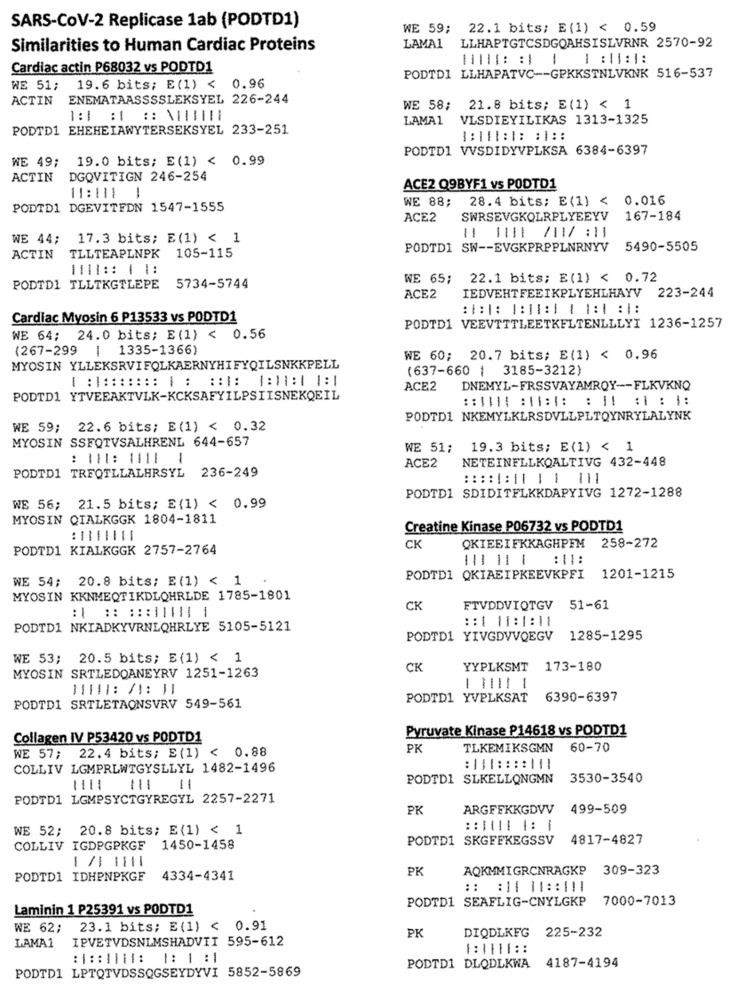
Examples of LALIGN-derived SARS-CoV-2 replicase protein (P0DTD2) similarities to human cardiac proteins. Vertical lines indicate identities between the amino acids; double dots indicate conserved amino acid substitutions. WE stands for Waterman-Eggert score (the larger the score, the less likely the similarity is due to chance). E is a measure of probability inverse to the WE score, so the smaller E, the less likely the match is due to chance. The numbers following the protein name are the UniProtKB identifiers. Numbers following the amino acid sequence (or in parentheses before the sequence) define the place of the match in the amino acid sequences of the protein.

**Figure 4 ijms-24-12177-f004:**
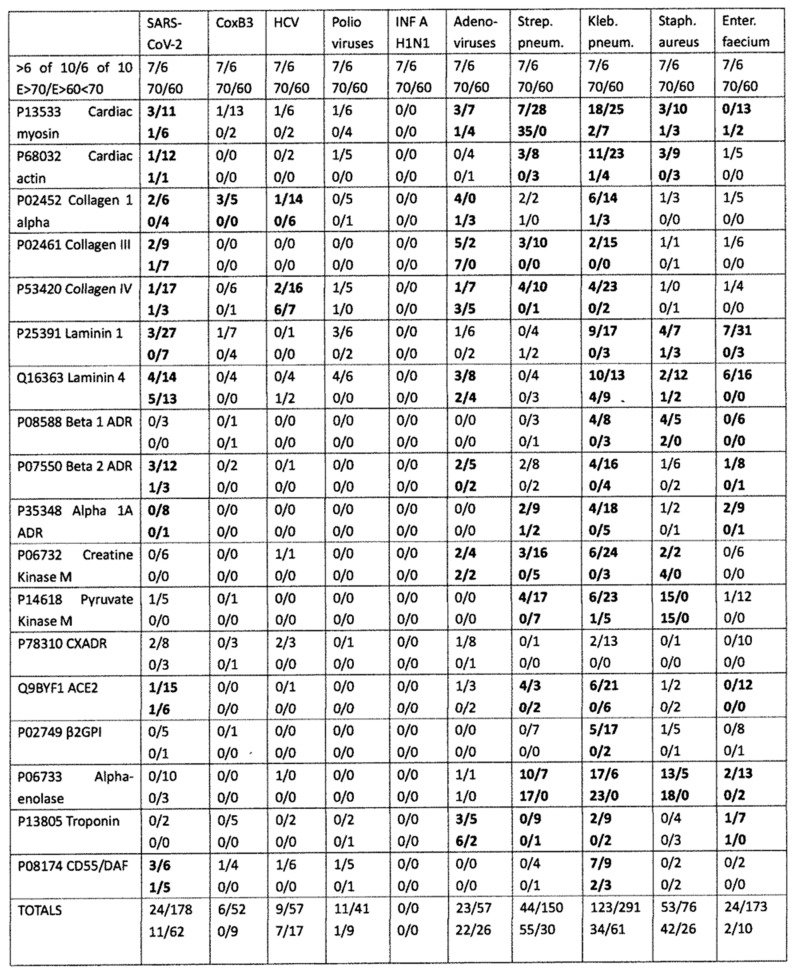
Summary of similarities between microbes associated with COVID-19 and human cardiac proteins. See Figure 1 for an explanation of the arrangement of numbers in each box. The bolded numbers are significantly different from the rest of the results.

**Figure 5 ijms-24-12177-f005:**
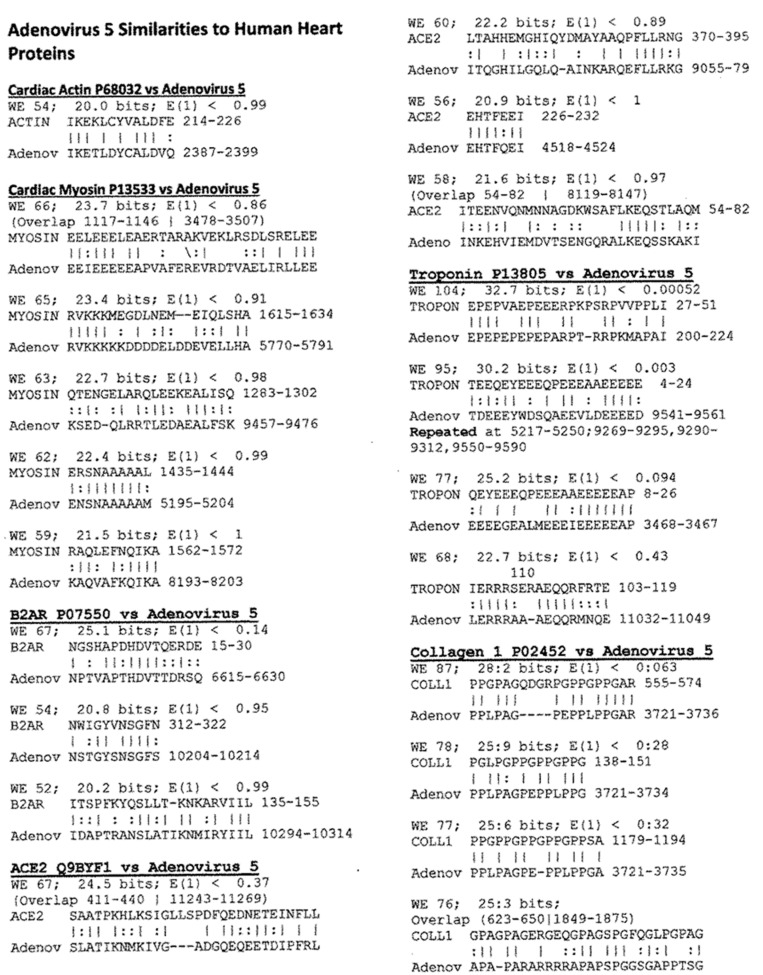
Examples of LALIGN-derived similarities between adenovirus type 5 and human cardiac proteins. WE stands for Waterman-Eggert score (the larger the score, the less likely the similarity is due to chance). E is a measure of probability inverse to the WE score, so the smaller E, the less likely the match is due to chance. The numbers following the protein name are the UniProtKB identifiers. Numbers following the sequences (or preceded by “overlap”) define the place of the sequence in the amino acid sequence of the protein.

**Figure 6 ijms-24-12177-f006:**
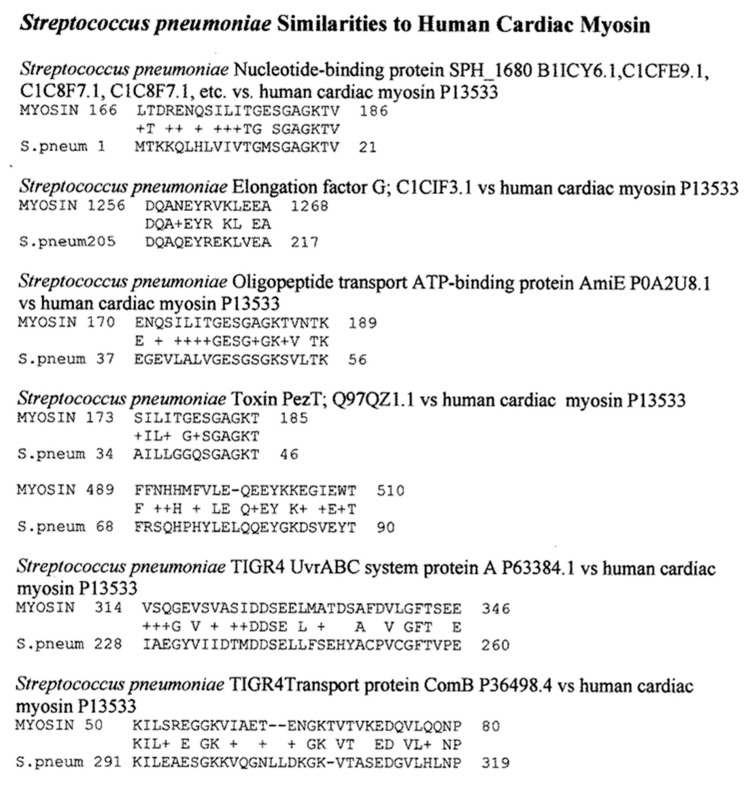
Examples of BLASTP-derived similarities between Streptococcus pneumoniae proteins and cardiac myosin. Plus signs (+) indicate conserved amino acid substitutions. Numbers following the protein name are the UniProtKB identifiers. The numbers before and after the sequences define the place of the sequence in the amino acid sequence of the protein.

**Figure 7 ijms-24-12177-f007:**
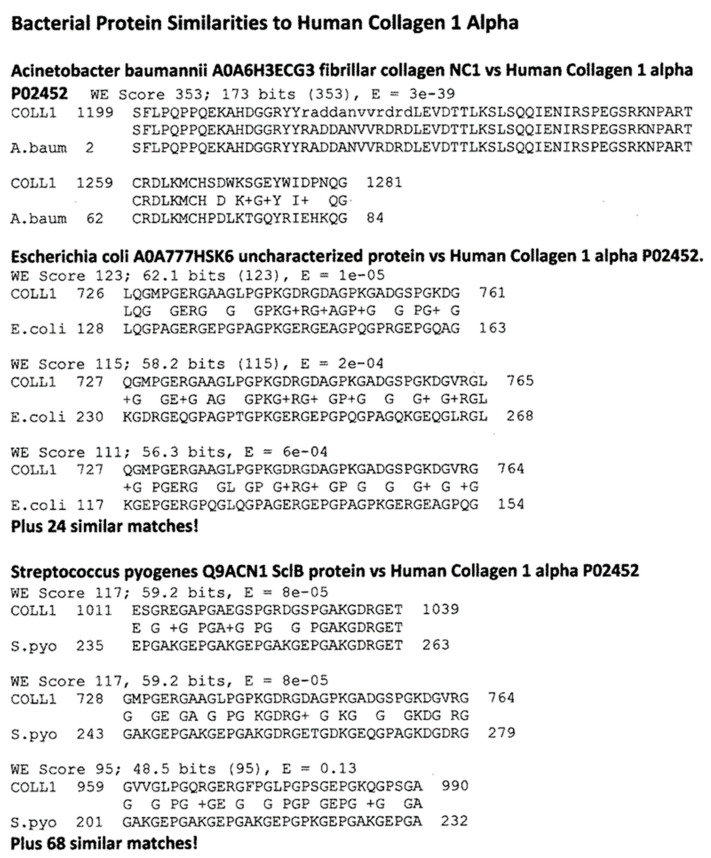
Examples of BLASTP-derived similarities between proteins of bacteria associated with COVID-19 collagen type 1 alpha. Plus signs (+) indicate conserved amino acid substitutions. Numbers following the protein name are the UniProtKB identifiers. The numbers before and after the sequences define the place of the sequence in the amino acid sequence of the protein.

**Figure 8 ijms-24-12177-f008:**
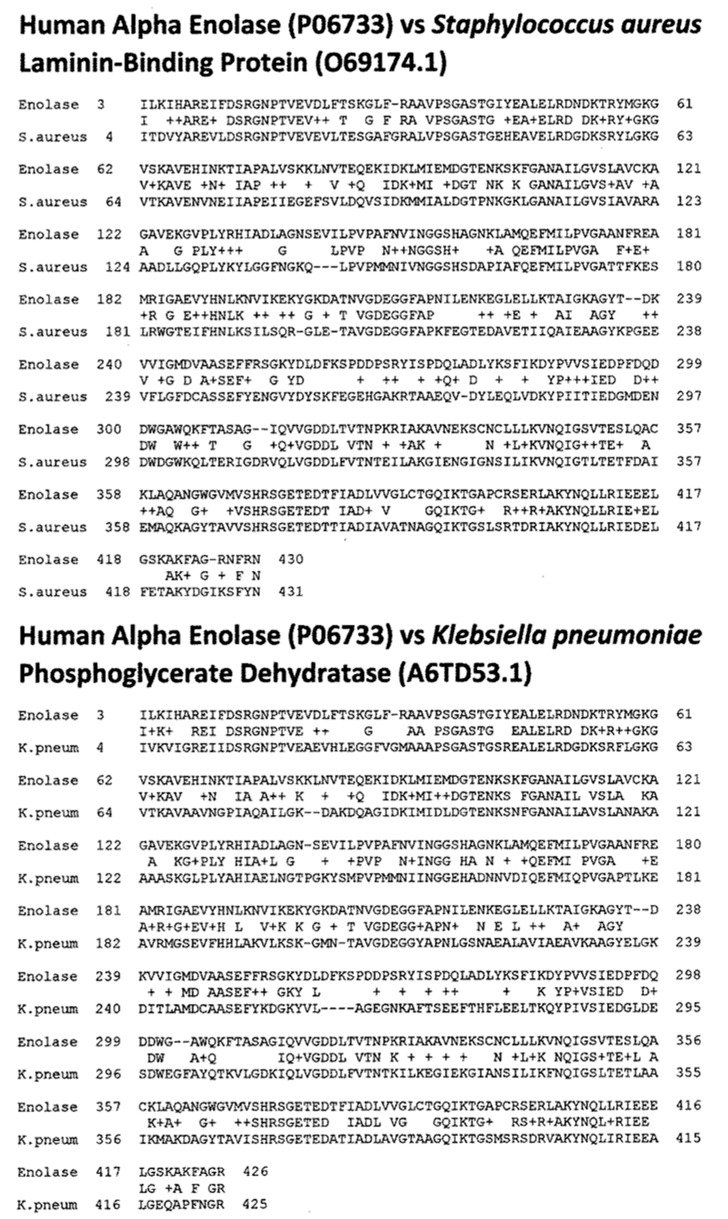
Examples of BLASTP-derived similarities between proteins of bacteria associated with COVID-19 and human alpha enolase. Actually, both proteins are phosphoglycerate dehydratases that also go by the name “laminin-binding protein”. Plus signs (+) indicate conserved amino acid substitutions. Numbers following the protein name are the UniProtKB identifiers. The numbers before and after the sequences define the place of the sequence in the amino acid sequence of the protein.

**Figure 9 ijms-24-12177-f009:**
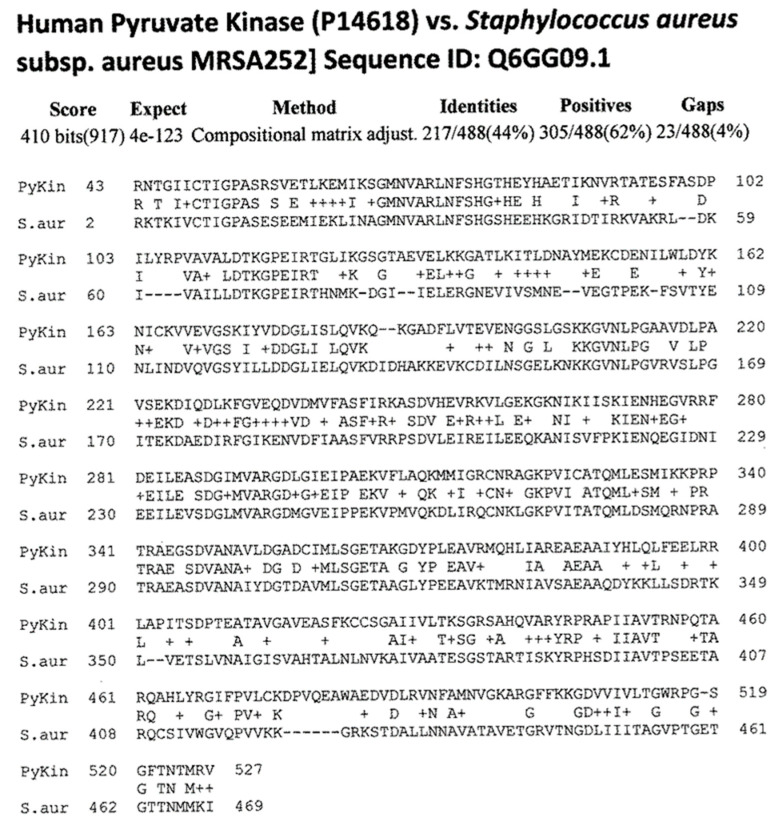
Example of BLASTP-derived similarities between proteins of methicillin-resistant Staphylococcus aureus and human pyruvate kinase. Plus signs (+) indicate conserved amino acid substitutions. Numbers following the protein name are the UniProtKB identifiers. The numbers before and after the sequences define the place of the sequence in the amino acid sequence of the protein.

**Figure 10 ijms-24-12177-f010:**
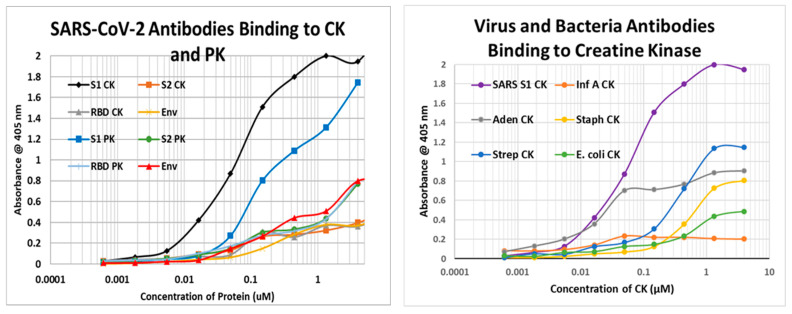
(**Left**) Quantitative ELISA binding curves for SARS-CoV-2 antibodies binding to creatine kinase (CK) and pyruvate kinase (PK). S1-CK = SARS-CoV-2 spike protein S1 fragment antibody binding to CK; S2 = SARS-CoV-2 spike protein fragment S2 antibody; RBD = SARS-CoV-2 spike protein RBD fragment antibody. (**Right**) Comparison of Influenza A antibody (Inf A), Adenovirus antibody (Aden), *Streptococcal* antibody (Strep), *Staphylococcal* antibody (Staph), and *Escherichia coli (E. coli)* antibody binding to creatine kinase (CK).

**Figure 11 ijms-24-12177-f011:**
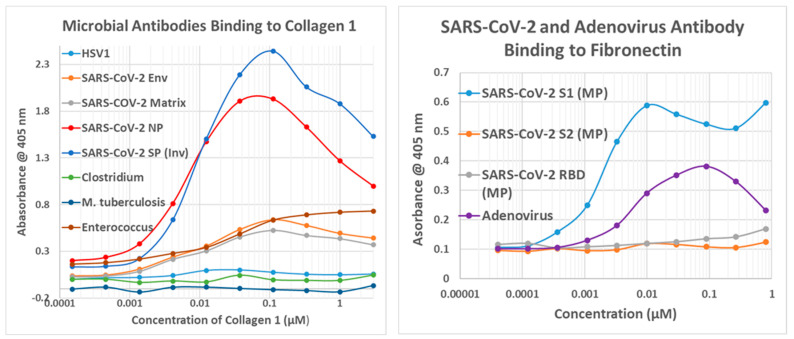
(**Left**) Quantitative ELISA results showing binding of SARS-CoV-2 envelope (Env), matrix, nucleoprotein (NP), and spike protein (SP) polyclonal antibodies binding to collagen type 1 (Collagen 1) compared with binding of some bacterial antibodies. (**Right**) SARS-CoV-2 spike protein fragment S1, S2, or RBD antibodies and adenovirus antibodies to fibronectin. MP = Millipore.

**Figure 12 ijms-24-12177-f012:**
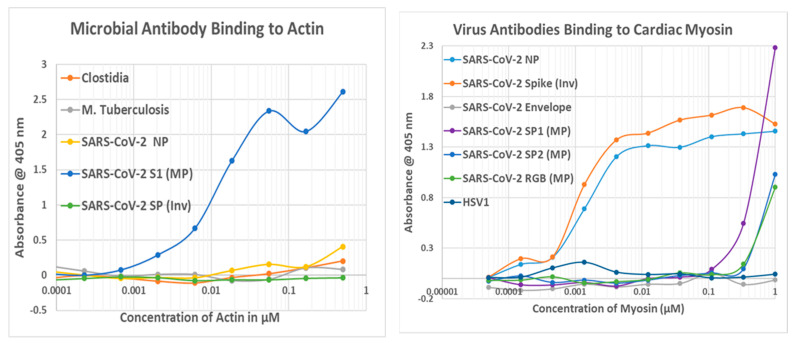
(**Left**) Quantitative ELISA results showing binding of SARS-CoV-2 nucleoprotein (NP), spike protein (SP), or spike protein S1 fragment (S1) antibodies to actin compared with antibodies against Clostridia and Mycobacterium tuberculosis. (**Right**) SARS-CoV-2 nucleoprotein (NP), spike protein fragment S1 (SP1), fragment S2 (SP2), or fragment RBD binding to cardiac myosin compared with human herpes simplex type 1 (HSV1) antibody binding. MP = Millipore; Inv = Invitrogen.

**Figure 13 ijms-24-12177-f013:**
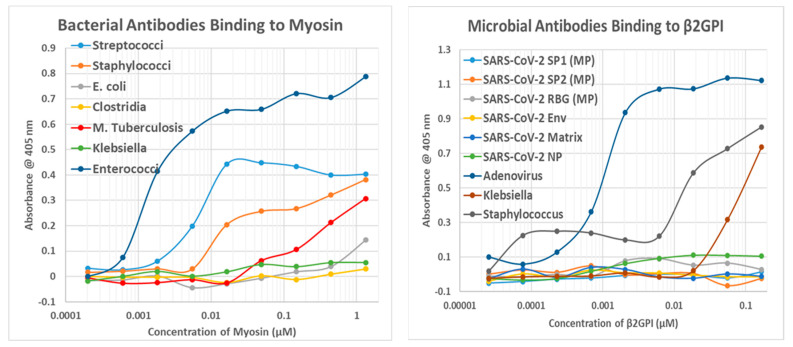
(**Left**) Results of quantitative ELISA experiments involving the binding of bacterial antibodies to myosin. *E. coli* = *Escherichia coli* antibody; *M. tuberculosis* = *Mycobacterium tuberculosis* antibody. (**Right**) Results of quantitative ELISA experiments involving binding of microbial antibodies to beta 2 glycoprotein I (β2GPI). The double curve for Staphylococcus binding may indicate both a high-affinity and a lower affinity binding site. SP1 = spike protein fragment 1; SP2 = spike protein fragment 2; RBD = spike protein RBD fragment; NP = nucleoprotein; MP = Millipore.

**Figure 14 ijms-24-12177-f014:**
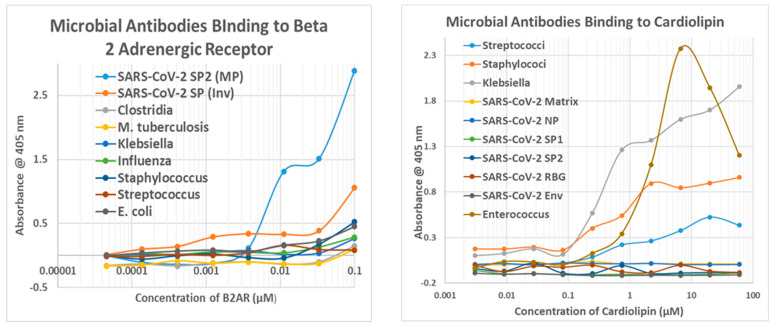
(**Left**) Results of quantitative ELISA experiments involving the binding of bacterial antibodies to the beta 2 adrenergic receptor (B2AR). *E. coli* = *Escherichia coli* antibody; *M. tuberculosis* = *Mycobacterium tuberculosis* antibody. SP = spike protein; SP1 = spike protein fragment 1; MP = Millipore; Inv = Invitrogen. (**Right**). Results of quantitative ELISA experiments involving the binding of microbial antibodies to cardiolipin. NP = nucleoprotein; SP1 = spike protein fragment 1; SP2 = spike protein fragment 2; RBG = spike protein RBG fragment; Env = envelope protein.

**Figure 15 ijms-24-12177-f015:**
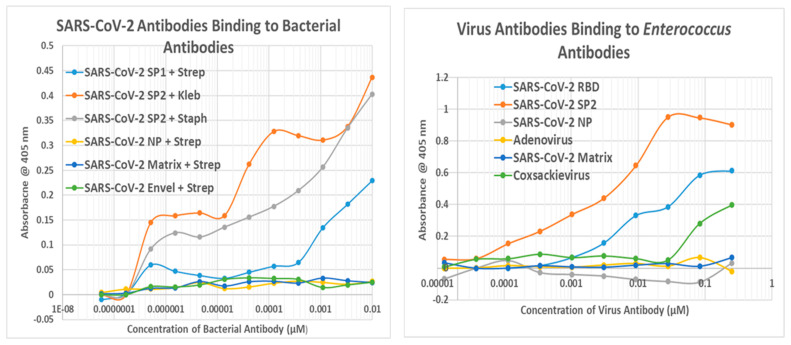
(**Left**) SARS-CoV-2 antibodies bind to Streptococcal (Strep), Staphylococcal (Staph), and Klebsiella (Kleb) antibodies. (**Right**) Enterococcus faecium antibodies binding to SARS-CoV-2, adenovirus, and coxsackievirus antibodies. SP1 = spike protein 1; SP2 = spike protein 2; NP = nucleoprotein; Envel = envelope protein.

**Figure 16 ijms-24-12177-f016:**
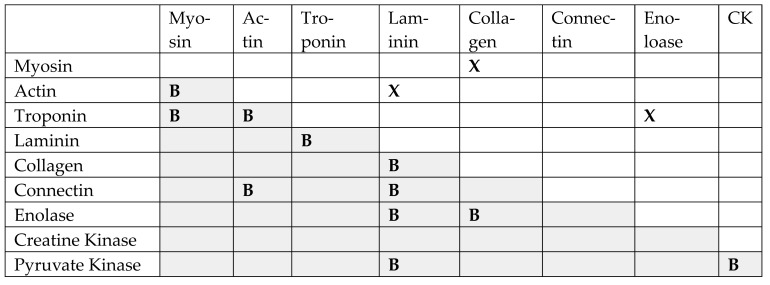
Table summarizing known binding (B) (gray background) or antibody cross-reactivity (X) (white background) among the permutations of proteins studied here. CK = creatine kinase See the text for references.

**Figure 20 ijms-24-12177-f020:**
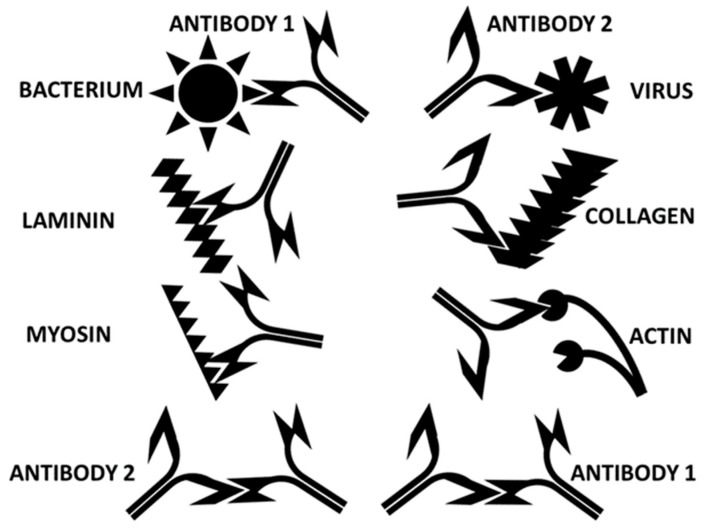
Cartoon illustrating some of the sets of antigen mimics that research has revealed are presented to the immune system in autoimmune cardiopathies. The virus presents antigens that mimic collagen and actin that elicit an antibody (“antibody 2”) that, in turn, mimics the complements of collagen and actin, which are laminin and myosin. The bacterium presents antigens that mimics, laminin and myosin, which results in another antibody (“antibody 1”) that mimics actin and collagen. Since the antigen sets are complementary, so are the antibodies. The result is total confusion about what is “self” and what is “non-self,” permitting autoimmune disease to progress. Note that the set of antigens listed here is not intended to be complete but merely illustrative. See the text for a discussion of additional sets of complementary sets of antigen mimics.

**Table 1 ijms-24-12177-t001:** A summary of known percentages of individuals diagnosed with antibodies against human cardiac autoantigens as a function of disease status or health. Numbers in brackets refer to article numbers in the references. Pre-COVID AM refers to autoimmune myocarditis diagnosed in patients prior to the COVID-19 pandemic; Pre-COVID DCM refers to patients diagnosed with dilated cardiomyopathy prior to the COVID-19 pandemic; # = number; Hosp. refers to hospitalized patients; ICU refers to patients admitted to an intensive care unit. The number of patients in some columns varied from row to row because the cited reference is a review aggregating several previous studies. AM = autoimmune myocarditis; DCM = dilated cardiomyopathy; Hosp. = hospitalized (severe); ICU = intensive care unit admitted; ACE2 = angiotensin converting enzyme 2; ANCA = perinuclear anti-neutrophil cytoplasmic antibody; β2GPI = anti-beta 2 glycoprotein I; β Adrenergic Rec. = beta adrenergic receptors 1 or 2; BPI = bactericidal permeability-inducing protein (vasculitis-associated).

Studies [36,37,38,39,40,41,42,43,44,45]	[42] Review	[42] Review & [50]	[46,50,51]	[47,49,51]	[47,48]	[42,48,51]
Antibody Targets in Autoimmune Cardiopathies	Pre-COVID AM	Pre-COVID DCM	Hospitalized COVID-19 Patients	ICU COVID-19 Patients	Out-Patient COVID-19	Healthy Controls
# of Patients Studied	Varied	Varied	147; 104; 118	29; 36; 118	24; 118	30; 36; Varied
Actin		71%				
ACE2			3.8–4.0%	27.2%		0.0%
ANCA				8.3–10.3%	8.3%	
β2GPI			41.6%	30.9%	4.0%	0.0%
β Adrenergic Rec.		30–75%				12–18%
BPI			23.5%			
Cardiolipin				20.6%	25.0%	3.0%
Collagen						
Enolase						
Laminin	73%	78%				6%
Mitochondria	91%	57%				0.0%
Myosin			7.9%			
Phospholipids			7.9%			
Tropomyosin		55%				
Troponin 1			47.8%			
Cardiac Muscle	59%	28–45%	68%	27.8%	0.0%	0–3.0%
Skeletal Muscle				19.4%	0.0%	
Smooth Muscle				30.6%	16.7%	

**Table 2 ijms-24-12177-t002:** Statistics comparing the significance of the total number of matches of SARS-CoV-2 (SARS) to the number of similarities to cardiac proteins from Figure 4 to those of control viruses: coxsackievirus type B3 (CoxB3); hepatitis C virus (HepC); poliovirus (polio); influenza A H1N1 (INFA); and adenovirus type 5 (Ad5). The total number of matches meeting the criterion of greater than 5 of 10 identical amino acids in a sequence of ten is provided under each virus name (“ID > 5 of 10”). A student’s paired T-test was used to compare the paired combinations. Because every possible combination was compared, a Bonferroni correction was applied such that for a difference to be significant, the *p* value needed to be less than 0.005 (indicated with a *).

	CoxB3 58	HCV 66	Polio 53	INFA 0	Ad5 80
SARS 202	t = 5.27 *p* < 0.0001 *	t = 4.16 *p* = 0.0007 *	t = 6.74 *p* < 0.0001 *	t = 7.09 *p* < 0.0001 *	t = 4.73 *p* = 0.0002 *
CoxB3 58		t = 0.38 *p* = 0.71	t = 0.48 *p* = 0.64	t = 3.53 *p* = 0.0026 *	t = 1.68 *p* = 0.11
HCV 66			t = 0.65 *p* = 0.52	t = 3.90 *p* = 0.0010 *	t = 0.86 *p* = 0.40
Polio 53				t = 3.42 *p* = 0.0032 *	t = 2.14 *p* = 0.05
INFA 0					t = 5.41 *p* < 0.0001 *

**Table 3 ijms-24-12177-t003:** Statistics comparing the significance of the number of high-quality SARS-CoV-2 (SARS) similarities to cardiac proteins to those of control viruses: coxsackievirus type B3 (CoxB3); hepatitis C virus (HepC); poliovirus (polio); influenza A H1N1 (INFA); and adenovirus type 5 (Ad5). The total number of matches that had an E score of 60 or greater (“E Score > 59”) is provided under each virus name. A student’s paired T-test was used to compare the paired combinations. Because every possible combination was compared, a Bonferroni correction was applied such that in order for a difference to be significant, the *p* value needed to be less than 0.005 (indicated with a *).

E Score >59	CoxB3 9	HCV 24	Polio 10	INFA 0	Ad5 48
SARS 73	t = 3.55 *p* = 0.0024 *	t = 2.39 *p* = 0.03	t = 3.53 *p* = 0.0026 *	t = 4.01 *p* = 0.0009 *	t = 1.32 *p* = 0.21
CoxB3 9		t = 1.05 *p* = 0.31	t = 0.0000 *p* = 1.00	t = 2.03 *p* = 0.06	t = 3.27 *p* = 0.0045 *
HCV 24			t = 1.03 *p* = 0.32	t = 1.71 *p* = 0.11	t = 2.14 *p* = 0.05
Polio 10				t = 2.60 *p* = 0.02	t = 3.47 *p* = 0.0029 *
INFA 0					t = 3.96 *p* = 0.0010 *

**Table 4 ijms-24-12177-t004:** Summary of quantitative ELISA experiments involving the binding of polyclonal antibodies against virus antigens to cardiac proteins and cardiolipin. Numbers are binding constants derived from the inflection points of the binding curves, examples are shown in Figure 11, Figure 12 and Figure 13. Coll 1 = collagen type 1; Coll IV = collagen type IV; Fibro = fibronectin; Inv = Invitrogen; Mab = monoclonal antibody (mouse); HSV1 = human herpes simplex virus type 1. * Human actin only; no binding to porcine actin.

SARS-CoV-2	Cardio-lipin	Myosin	Actin	Coll I	Coll IV	Laminin	Fibro	B2AR	Creatine Kinase	Pyruvate Kinase
Spike 1 (Millipore)	>100 µM	220 nM	>1 µM	8 nM	10 nM	1 µM	>1 µM	100 nM	60 nM	300 nM
Spike Protein (Inv)	>100 µM	10 nM	>1 µM	>1 µM	>1 µM	>1 µM	2 nM	300 nM	--	--
S2 (Millipore)	>100 µM	>1 µM	>1 µM	>1 µM	>1 µM	>1 µM	>1 µM	50 nM	>10 µM	>1 µM
RBD (Millipore)	>100 µM	>1 µM	>1 µM	4 nM	>1 µM	10 nM	>1 µM	>1 µM	>10 µM	>1 µM
Envelope (Inv)	>100 µM	>1 µM	>1 µM	>1 µM	>1 µM	>1 µM	>1 µM	>1 µM	>10 µM	>1 µM
Matrix (Inv)	>100 µM	300 nM	>1 µM	>1 µM	>1 µM	>1 µM	>1 µM	>1 µM	>10 µM	>1 µM
Nucleoprotein (Inv)	>100 µM	>1 µM	>1 µM	5 nM	20 nM	>1 µM	>1 µM	>1 µM	700 nM	400 nM
OTHER VIRUSES										
Adenovirus	>100 µM	7 nM	>1 µM	>1 µM	>1 µM	2 nM	8 nM	>1 µM	1.5 nM	>1 µM
Influenza A	>100 µM	50 nM	>1 µM	>1 µM	2 nM	>1 µM	>1 µM	>1 µM	>10 µM	>1 µM
Coxsackie B MAb mix	>100 µM	50 nM	>1 µM	>1 µM	>1 µM	100 nM	>1 µM	>1 µM	--	--
Coxsackie B3 Monkey	>100 µM	30 nM	0.7 nM *	>1 µM	2 nM	>1 µM	0.8 nM	>1 µM	>10 µM	1 µM
HSV1	>100 µM	>1 µM	>1 µM	>1 µM	>1 µM	>1 µM	>1 µM	>1 µM	>10 µM	100 nM

**Table 5 ijms-24-12177-t005:** Physiological concentrations of cardiac proteins [75,76,77,78,79,80,81,82,83,84,85,86,87,88,89,90]. The numbers in brackets refer to the references. * = healthy heart but micromolar actin found in severe lung injury, sepsis, etc. [89,90]; ^ = range from healthy (0) to severe myocarditis (250 nM) [86]; Conc. = concentration; # Calculated as follows: 5000 units/mg pyruvate kinase (PK) activity [87] and 12–46 U/mL in serum from heart failure patients [88], and molecular weight (MW) if PK is 57 kD/chain. So 50 U/mL = 0.2 mg/mL = 0.2 uM; is ~ calculated from total hydroxyproline in the heart, which is 3 mg/gram of tissue, and the MW of collagen is 300 kD, so 3 mg/g = 10 µM [79], and collagen I makes up 80% of heart collagen; collagen III, 11%; and collagen IV, about 5% [80].

	Significant K_D_ Found Here	Normal Serum Conc.	Heart Conc.	Source
Actin	10 nM	0 *	2 µM	[75]
Beta 2 Adrenergic Receptor	50 nM	n/r	15 nM	[76]
Cardiolipin	400 nM–2 µM	10 µM	1.6 µM	[77,78]
Collagen I Fibrillar	4–8 nM	n/r	8 µM ~	[79,80]
Collagen III Fibrillar	--	n/r	1 µM ~	[79,80]
Collagen IV Basement Membrane	2–20 nM	n/r	0.5 µM ~	[79,80]
Creatine Kinase	60 nM	20–50 nM	40–60 nM	[81,82]
Fibronectin	0.3–100 nM	1–3 µM	0.5 µM	[83,84]
Laminins	0.8–100 nM	n/r	0.2–0.4 µM	[85]
Myosin	3–50 nM	0–250 nM ^	1 µM	[82,86]
Pyruvate Kinase	100 nM	50–200 nM #	50–200 nM #	[87,88]

**Table 6 ijms-24-12177-t006:** Summary of quantitative ELISA experiments involving the binding of polyclonal and monoclonal antibodies against bacterial antigens to cardiac proteins and cardiolipin. Numbers are binding constants derived from the inflection points of the binding curves, examples are shown in Figure 11(Right), Figure 12(Left), Figure 13(Left), Figure 15 and Figure 16. Coll 1 = collagen type 1; Coll IV = collagen type IV; Fibro = fibronectin; Inv = Invitrogen; Mab = monoclonal antibody (mouse); *S. aureus* = *Staphylococcus aureus*; GAS = group A *Streptococci*; Strep = Streptococcus pneumoniae; *E. coli* = *Escherichia coli*; *M. tuberculosis* = *Mycobacterium tuberculosis*; Mab = monoclonal antibody; the numbers following Mab identify the specific clone (see Methods and Materials).

Bacteria	Cardio-Lipin	Myosin	Actin	Coll I	Coll IV	Laminin	Fibro	B2AR	Creatine Kinase	Pyruvate Kinase
*S. aureus*	2 µM	15 nM	>1 µM	>1 µM	22 nM	>1 µM	6 nM	>1 µM	500 nM	>10 µM
*GAS (Rabbit)*	1 µM	8 nM	>1 µM	>1 µM	>1 µM	0.8 nM	>1 µM	>1 µM	300 nM	2 µM
*Strep (Goat)*	4 µM	10 nM	>1 µM	>1 µM	>1 µM	>1 µM	--	>1 µM	>10 µM	>1 µM
*GAS MAb 1-10698*	>100 µM	0.2 nM	>1 µM	4 nM	100 nM	2 nM	>1 µM	--	>10 µM	4 µM
*GAS MAb 1-10700*	>100 µM	3 nM	>1 µM	>1 µM	300 nM	>1 µM	100 nM	--	>10 µM	>10 µM
*Klebsiella*	400 nM	>1 µM	>1 µM	>1 µM	>1 µM	>1 µM	>1 µM	>1 µM	>10 µM	>10 µM
*E. coli*	1 µM	>1 µM	>1 µM	>1 µM	>1 µM	>1 µM	>1 µM	>1 µM	>10 µM	>1 µM
*Clostridium*	>100 µM	>1 µM	>1 µM	>1 µM	>1 µM	>1 µM	0.3 nM	>1 µM		
*M. tuberculosis*	>100 µM	>1 µM	>1 µM	>1 µM	>1 µM	>1 µM	0.8 nM	>1 µM	>10 µM	2 µM
*Enterococcus*	2 µM	1.3 nM	>1 µM	30 nM	>1 µM	>1 µM	--	--	>10 µM	4 µM

**Table 7 ijms-24-12177-t007:** Summary of virus antibodies binding to bacterial antibodies by double-antibody ELISA. Pairs of antibodies that bind to each other with significant affinity are bolded for ease of identification. Entero. = *Enterococcus;* Strep. pneum. = *Streptococcus pneumoniae*; GAS = Group A *Streptococci*; Staph = *Staphylococcus aureus*; *E. coli* = *Escherichia coli*; Clost. = *Clostridium*; M. tb. = *Mycobacterium tuberculosis*; HRP = horse radish peroxidase-labeled antibody; Gt = goat antibody; Rab = rabbit antibody; Ms = mouse monoclonal antibody; S1 = SARS-CoV-2 spike protein S1 region; S2 = SARS-CoV-2 spike protein S2 region; RBD = SARS-CoV-2 spike protein RBD region; MP = Millipore; Inv = Invitrogen; CVB = coxsackievirus B1-B6; HSV = human herpes simplex virus; CMV = cytomegalovirus.

SARS-CoV-2	*Entero.* HRP	*S. pneum.* Gt HRP	*GAS* Rab HRP	*Staph* Gt HRP	*Kleb* Gt HRP	*E. coli* Gt HRP	*Clost* Rab HRP	*M. tb* GP
S1 (MP) Rab	250 nM	0.1 nM	1.7 nM	1 nM	>1 µM	>1 µM	>1 µM	NP
S2 (MP) Rab	3 nM	0.2 nM	1.8 nM	36 nM	0.2 nM	>1 µM	>1 µM	NP
RBD (MP) Rab	8 nM	0.4 nM	4.8 nM	>1 µM	>1 µM	>1 µM	>1 µM	NP
Envelope Rab	>1 µM	>1 µM	>1 µM	>1 µM	>1 µM	>1 µM	>1 µM	NP
Matrix Rab	>1 µM	>1 µM	>1 µM	>1 µM	>1 µM	>1 µM	>1 µM	NP
Nucleocapsid Rab	>1 µM	>1 µM	>1 µM	>1 µM	>1 µM	>1 µM	>1 µM	NP
OTHER VIRUSES								
Adenovirus Gt	>1 µM	>1 µM	>1 µM	>1 µM	>1 µM	NP	>1 µM	>1 µM
Influenza A Gt	>1 µM	>1 µM	>1 µM	>1 µM	>1 µM	>1 µM	>1 µM	>1 µM
CVB blend Ms	--	0.2 nM	>1 µM	120 nM	>1 µM	0.3 nM	120 nM	>1 µM
CVB3 monkey	60 nM	500 nM	20 nM	10 nM	--	10 nM	1.5 nM	10 nM
HSV1 Gt	>1 µM	>1 µM	>1 µM	24 nM	>1 µM	>1 µM	>1 µM	>1 µM
HSV2 Gt	--	>1 µM	>1 µM	>1 µM	--	>1 µM	--	>1 µM
CMV Rab	--	>1 µM	>1 µM	>1 µM	--	>1 µM	--	70 nm

**Table 8 ijms-24-12177-t008:** Summary of the proteomic similarity data and the ELISA results for the cardiac proteins in this study. The proteomic data are from Figure 2, where the formalism is explained. Boxes in orange are those for which significant microbial antibody binding was observed to the cardiac protein. White boxes are those for which no significant binding was observed. ND means that the proteomic search was not performed, so no prediction concerning binding was made. For cardiolipin, the negative signs (-) indicate no binding was observed; the plus signs (+) indicate that significant binding was observed. CoxB3 = coxsackievirus B3; InfA H1N1 = influenza virus H1N1; Strep. = Group A *Streptococci*; Staph. = *Staphylococcus*; Enter. = *Enterococcus;* Kleb. = *Klebsiella*.

	SARS-CoV-2 Spike Protein	SARS-CoV-2 Whole Virus	CoxB3	INF A H1N1	Adenovirus Type 5	*Strep. pneum.*	*Staph.* *aureus*	*Enter.* *faecium*	*Kleb. pneum.*
>6 of 10/6 of 10 E > 70/E > 60 < 70	7/6 70/60	7/6 70/60	7/6 70/60	7/6 70/60	7/6 70/60	7/6 70/60	7/6 70/60	7/6 70/60	7/6 70/60
P13533 Cardiac myosin	1/0 0/0	3/11 1/6	1/13 0/2	0/0 0/0	3/7 1/4	7/28 35/0	3/10 1/3	0/13 1/2	18/25 2/7
P68032 Cardiac actin	0/4 0/0	1/12 1/1	0/0 0/0	0/0 0/0	0/4 0/1	3/8 0/3	3/9 0/3	1/5 0/0	11/23 1/4
P02452 Collagen 1 alpha	1/1 0/1	2/6 0/4	3/5 0/0	0/0 0/0	0/0 0/0	2/2 1/0	1/3 0/0	1/5 0/0	6/14 1/3
P53420 Collagen IV	0/1 0/0	1/17 1/3	0/6 0/1	0/0 0/0	0/0 0/0	4/10 0/1	1/0 0/1	1/4 0/0	4/23 0/2
P25391 Laminin 1	1/7 0/1	3/27 0/7	1/7 0/4	0/0 0/0	1/6 0/2	0/4 1/2	4/7 1/3	7/31 0/3	9/17 0/3
P07550 Beta 2 ADR	0/4 0/1	3/12 1/3	0/2 0/0	0/0 0/0	2/5 0/2	2/8 0/2	1/6 0/2	1/8 0/1	4/16 0/4
P06732 Creatine Kinase	0/1 0/0	0/6 0/0	0/0 0/0	0/0 0/0	2/4 2/2	3/16 0/5	2/2 4/0	0/6 0/0	6/24 0/3
P14618 Pyruvate Kinase	0/0 0/0	1/5 0/0	0/1 0/0	0/0 0/0	0/0 0/0	4/17 0/7	15/0 15/0	1/12 0/0	6/23 1/5
P02749 β2GPI	0/1 0/0	0/5 0/1	0/1 0/0	0/0 0/0	0/0 0/0	0/7 0/0	1/5 0/1	0/8 0/1	5/17 0/2
Cardiolipin	--	--	--	--	--	+	+	+	+
Fibronectin	ND	ND	ND	ND	ND	ND	ND	ND	ND

**Table 9 ijms-24-12177-t009:** Summary of studies comparing antibody reactivity to human cardiac proteins among autoimmune myocarditis (AM) and dilated cardiomyopathy (DM) pre-COVID-19 (condensed from Table 1); hospitalized and intensive care unit (ICU) admitted COVID-19 patients (condensed from Table 1); rabbit polyclonal and human monoclonal antibodies (Ab) against the SARS-CoV-2 spike protein (SP) or non-SP proteins; and the results of this study regarding SP, non-SP, and bacterial antibodies. The numbers in brackets refer to the References. Numbers followed by a percent sign (%) are the percent of patients in the group that displayed antibodies against that antibody. + signs indicate that antibodies are known to cross-react with that protein, but no data exists as to what percentage of patients exhibit these antibodies. The stars (*) indicate that while antibody tests were not carried out on enolase cross-reactivity in our study, our proteomic results demonstrate no significant similarities of enolase to SARS-CoV-2 proteins but very extensive and high-quality ones to all of the bacteria studied, and these have been independently verified for several of the bacteria [119,120,121]. Given the failure of any animal or human SARS-CoV-2 antibodies to cross-react with enolase, it is therefore conjectured that the source of these antibodies found in myocarditis patients must be bacterial. The carrot (^) indicates that bacteria are known to induce anti-mitochondrial antibodies [122,123], and SARS-CoV-2 proteins are often produced using recombinant DNA techniques in *E. coli*, which has been demonstrated to result in contamination of the viral protein with bacterial antigens [124,125]; The possibility, therefore, must be considered that cross-reactivity to mitochondrial antigens by antibodies putatively induced by SARS-CoV-2 antigens is due to bacterial contaminants.

Antibody Target	Pre- COVID AM & DCM	Severe COVID-19 Hospitalized & ICU	Human Mab SARS-CoV-2 SP	Rabbit SARS-CoV-2 SP Ab	Rabbit SARS-CoV-2 SP Ab	Human Mab Non-SP SARS-CoV-2	Rabbit Non-SP SARS-CoV-2	Non-SP SARS-CoV-2 Abs	Poly-clonal Bacterial Abs
STUDIES [36,37,38,39,40,41,42,43,44,45]	[42,50,98,99,100,101,102,103,104,105,106,107,108,109,110,111,112,113,114,115,116]	[46,47,48,49,50,51,98,99,100,101,102,103,104,105,106,107,108,109,110,111,112,113,114,115,116]	[118]	[117,118]	This Study	[118]	[117,118]	This Study	This Study
Actin	71%	+	+	+	+	+	+	-	-
ACE2	+	3.8–27.2%							
ANCA	+	8.3–10.3%							
β2GPI	+	30.9–41.6%	-	-	-	-		-	+++
Adren. Rec.	30–75%	+			+			-	-
Cardiolipin	+	20.6%			-			-	+++++
Collagen	+	+	+	+	+	+	+	-	+++
CK	+	+			+			-	+
Enolase	+		-	-	-	-			***
Fibronectin	+	+	-	-	-	-		-	++++
Laminin	73–78%	+	-	-	+	-		-	++
Mitochondria	57–91%	+	+ ^	+ ^		+ ^			
Myosin	+	7.9%	+	+	+	-	+	+	++++
Phospholipids	+	7.9%	+	+		-	+		
PK	+	+			-			-	+++
Tropomyosin	55%	+	-	-		+	+		
Troponin 1	+	47.8%							
Cardiac Muscle	28–59%	27.8–68%							
Skeletal Muscle		19.4%							
Smooth Muscle		30.6%							

**Table 10 ijms-24-12177-t010:** List of antigens utilized in experiments. # = number.

Product Name	Species	Supplier	Product #	Purity
Actin, alpha cardiac	Human	Hypermol	8201-02	>99%
Actin from bovine muscle	Cow	Sigma-Aldrich	A3653	>90%
Actin from rabbit muscle	Rabbit	Sigma-Aldrich	A2522	>85%
Cardiolipin from heart	Cow	Sigma-Aldrich	C0563	>97%
Collagen Type I from placenta	Human	Sigma-Aldrich	C7774	>95%
Collagen Type IV from placenta	Human	Sigma-Aldrich	C7521	>95%
Collagen Type IV	Mouse	Cultrex	3410-010-01	>99%
Creatine Kinase (cardiac MM)	Human	Sigma-Aldrich	C9858-100UN	>1000 U/mg
Fibronectin from plasma	Human	Cultrex	3420-001-01	>99%
Fibronectin from plasma	Human	Sigma-Aldrich	F2006	>85%
Laminin from fibroblasts	Human	Sigma-Aldrich	L4544	>90%
Laminin-1	Mouse	Cultrex	3400-010-01	>99%
Myosin light chain, cardiac	Human	Sigma-Aldrich	M4824	>90%
Myosin Ca^2+^ Activated, cardiac	Pig	Sigma-Aldrich	M0531	0.5 U/mg
Pyruvate Kinase (recombinant)	Human	Sigma-Aldrich	SAE0021	>100 U/µg

**Table 11 ijms-24-12177-t011:** List of antibodies utilized in experiments. # = number.

Product Name	Species	Supplier	Product #
Actin antibody	Rabbit	Sigma-Aldrich	A2668
Adenovirus	Goat	Millipore	AB1056
*Clostridia*	Rabbit	Invitrogen	PA1-7210
Clostridium sp. HRP	Rabbit	US Biological	C5853-25C
Collagen IV	Rabbit	Novus Biologicals	NB120-6586
Coxsackie Virus B1-B6 Blend	Mouse	Millipore	MAB9410
*Enterococcus* HRP	Rabbit	Invitrogen	PA1-73122
*Escherichia coli*	Goat	abcam	AB13627
Goat Anti-Mouse IgG HRP	Goat	Sigma-Aldrich	A9917
Goat Anti-Rabbit IgG HRP	Goat	Invitrogen	65-6120
Herpes Simplex Virus Type 1	Goat	Invitrogen	PA1-7493
Influenza A HRP	Goat	Biodesign International	B65243G
*Klebsiella pneumoniae* HRP	Rabbit	Invitrogen	PA1-73176
Laminin antibody	Rabbit	Sigma-Aldrich	L9393
*Mycobacterium tuberculosis*	Rabbit	ABD Serotec	OBT0947
*Mycobacterium tuberculosis*	Guinea Pig	MyBioSource	MBS315001
Myosin antibody	Goat	Santa Cruz Biotechnology	12117
Rabbit Anti-Goat IgG HRP	Rabbit	Millipore	AP106P
Rabbit Anti-Guinea Pig HRP	Rabbit	abcam	AB6771
SARS-CoV-2 Envelope protein	Rabbit	Invitrogen	PA1-41158
SARS-CoV-2 Matrix protein	Rabbit	Invitrogen	PA1-41160
SARS-CoV-2 Nucleocapsid	Rabbit	Invitrogen	PA5-116894
SARS-CoV-2 Spike Protein RBD	Rabbit	Millipore	ABF1064
SARS-CoV-2 Spike Protein S1	Rabbit	Millipore	ABF1065
SARS-CoV-2 Spike Protein S1	Rabbit	Invitrogen	PA5-116916
SARS-CoV-2 Spike Protein S2	Rabbit	Millipore	ABF1063
*Staphylococcus aureus*	Rabbit	Invitrogen	PA1-7246
*Staphylococcus aureus* HRP	Rabbit	Invitrogen	PA1-73173
*Streptococcus* Group A	Goat	Invitrogen	PA1-7249
*Streptococcus* Group A HRP	Rabbit	Acris Antibodies	BP2026HRP
*Streptococcus pneumoniae*	Rabbit	Biodesign International	B65831R
*Streptococcus pneumoniae*	Rabbit	Invitrogen	PA1-7259

## Data Availability

Data not already provided in the manuscript can be obtained by contacting the primary investigator (RRB: rootbern@msu.edu).

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
