# Peer review of "SARS-CoV-2 and Its Bacterial Co- or Super-Infections Synergize to Trigger COVID-19 Autoimmune Cardiopathies"

_ijms, 2023, doi:10.3390/ijms241512177_

Round 1

Reviewer 1 Report

Dear authors, your manuscript is very detailed and complex, however before we can go further into details , there are a few major issues that have to be addressed in this review. 

Since this is such a long manuscript, I found it particularly difficult to read because the methods and materials section is at the end of the manuscript. Perhaps this is an editing mistake which must be addressed before any further reviewing. Another issue I had were the links within the manuscript such as from lines 848 to 852, these copy pasted links litter the manuscript and I wonder why were these included ? 

One other major issue is that the so called figures like figures 4 5 7 for example appear to be scanned tables which are pasted in the manuscript without even being adjusted to be straight, not to mention that these should be transformed into tables and not be scanned. 

Also the methods and materials section is quite vast and I find it rather difficult to interpret the time frame and sequence of the experiments as well as the author's choice of materials, these issues should be made much clearer. 

This manuscript shows potential and I do have several more comments on the experiments and study, before I can go into the details, the body, text and figures must be checked and properly edited as  the authors should address the abovementioned concerns. In the current form I find the manuscript disorganized and difficult to read and interpret. 

Rather good English. 

Author Response

REVEWER 1

Open Review

Quality of English Language

( ) I am not qualified to assess the quality of English in this paper
( ) English very difficult to understand/incomprehensible
( ) Extensive editing of English language required
( ) Moderate editing of English language required
(x) Minor editing of English language required
( ) English language fine. No issues detected

Yes

Can be improved

Must be improved

Not applicable

Does the introduction provide sufficient background and include all relevant references?

( )

(x)

( )

( )

Are all the cited references relevant to the research?

( )

(x)

( )

( )

Is the research design appropriate?

( )

(x)

( )

( )

Are the methods adequately described?

( )

(x)

( )

( )

Are the results clearly presented?

( )

(x)

( )

( )

Are the conclusions supported by the results?

( )

(x)

( )

( )

Comments and Suggestions for Authors

Dear authors, your manuscript is very detailed and complex, however before we can go further into details , there are a few major issues that have to be addressed in this review. 

Since this is such a long manuscript, I found it particularly difficult to read because the methods and materials section is at the end of the manuscript. Perhaps this is an editing mistake which must be addressed before any further reviewing.

THIS REVIEWER HAS APPARENTLY FAILED TO READ THE GUIDE TO AUTHORS OR LOOK AT THE SUBMISSION TEMPLATE: IJMS REQUIRES MATERIALS AND METHODS TO FOLLOW THE DISCUSSION. THIS ISN’T SOMETHING WE HAVE ANY CONTROL OVER.

Another issue I had were the links within the manuscript such as from lines 848 to 852, these copy pasted links litter the manuscript and I wonder why were these included ? 

THESE LINKS ARE INCLUDED BECAUSE THEY ARE TO WEBSITES WITH DATA SETS OR TOOLS THAT WERE USED DURING OUR RESEARCH. SINCE THEYARE NOT PUBLICATIONS, THEY DO NOT FIT REASONABLY INTO THE REFERENCES. THUS,  IT MADE SENSE TO INTRODUCE THEM WHERE WE FIRST USED THEM. WE HAVE DONE THIS IN PREVIOUS PAPERS WE HAVE PUBLISHED WITH IJMS. IN ANY CASE, THIS IS ONCE AGAIN AN EDITORIAL ISSUE, NOT A REVIEWER ISSUE.

One other major issue is that the so called figures like figures 4 5 7 for example appear to be scanned tables which are pasted in the manuscript without even being adjusted to be straight, not to mention that these should be transformed into tables and not be scanned. 

YES, THESE ARE SCANNED IMAGES. THEY ARE SCANNED BECAUSE OUR PREVIOUS EXPERIENCES WITH IJMS IS THAT NO MATTER HOW WE LAY OUT A TABLE, IT IS REFORMATTED BY THE EDITORS DURING PROOFING. MOREOVER, IF ONE READS THE GUIDE TO AUTHORS, ONE FINDS THAT IT IS NOT PERMITTED TO USE FONT MODIFICATIONS, ITALICS, COLORS, ETC. IN TABLES. WE, HOWEVER, REQUIRE THAT THE DATA BE LAID OUT THE WAY WE HAVE IT IN THESE FIGURES IN ORDER TO PRESENT THE DATA IN THE CLEAREST AND MOST CONCISE MANNER AND THEREFORE PRESENT THEM AS “FIGURES” INSTEAD OF “TABLES” IN ORDER TO MAKE SURE THE FORMATTING IS RETAINED. IN ANY EVENT, THIS IS ONCE AGAIN AN ISSUE FOR THE EDITORS, NOT FOR THE REVIEWER.

Also the methods and materials section is quite vast and I find it rather difficult to interpret the time frame and sequence of the experiments as well as the author's choice of materials, these issues should be made much clearer. 

HOW DOES THE TIME FRAME OR ORDER IN WHICH THE EXPERIMENTS WERE PERFORMED MAKE ANY DIFFERENCE TO THE RESULTS?

WE HAVE ADDED EXPLANATIONS OF OUR CHOICE OF MATERIALS IN THE MATERIALS AND METHODS SECTION, HIGHLIGHTED IN YELLOW.

This manuscript shows potential and I do have several more comments on the experiments and study, before I can go into the details, the body, text and figures must be checked and properly edited as  the authors should address the abovementioned concerns. In the current form I find the manuscript disorganized and difficult to read and interpret. 

INTERESTING, SINCE THE OTHER REVIEWER FOUND THE LOGIC AND PRESENTATION “well-composed, and the findings are clearly laid out.” HOWEVER, WHAT WE HAVE DONE, SINCE THE METHODS AND MATERIALS FOLLOW THE DISCUSSION IS TO ADD SOME BRIEF DESCRIPTIONS OF THE EXPERIMENTAL PROTOCOLS AT APPROPRIATE POINTS IN THE PRESENTATION OF THE RESULTS. THESE ARE HIGHLIGHTED IN YELLOW. WE HOPE THIS HELPS.

Comments on the Quality of English Language

Rather good English. 

THANK YOU.

Submission Date

08 July 2023

Date of this review

15 Jul 2023 14:09:05

Reviewer 2 Report

Ijms-2522608

This intriguing study investigates whether antibodies to SARS-CoV-2 mirror cardiac antigens and if such resemblance is adequate to justify the risk of autoimmune myocarditis associated with COVID-19 and its vaccination. It also considers the potential role of bacterial co-infections, both as incidental infections or as active instigators of autoimmune myocarditis. The research reveals that some antibodies against SARS-CoV-2 proteins, especially those that target elements of the spike protein, show cross-reactivity with certain cardiac proteins. The study underscores that SARS-CoV-2 proteins mimic human cardiac proteins to a significantly larger extent than other viruses, such as Ad5, coxsackieviruses, or HCV, which have a known risk of autoimmune heart diseases, or the influenza A H1N1 virus, which doesn't. The information about adenovirus is particularly noteworthy, given that Ad5 serves as a delivery mechanism for some of the SARS-CoV-2 spike protein vaccines currently in development. The manuscript is well-composed, and the findings are clearly laid out. However, some changes are suggested to enhance the manuscript's quality:

  1. The current title "COVID-19 Autoimmune Cardiopathies May Result from Complementary Antigens that Mimic Cardiac Proteins Expressed by Both SARS-CoV-2 and Bacterial Co- or Super-Infections" could be simplified for easier comprehension of the study's main points.

  1. The reference formatting should be made consistent throughout the manuscript. For instance, citations 70-73 were underlined while others were not. There were also inconsistencies in the citation format as seen in 188/189.

  1. The introduction to the LALIGN and Waterman-Eggert score could be moved to the introduction/results section.

  1. The ELISA quantitative analysis visuals appear to be in draft form and need to be refined, particularly with regard to text size and formatting.

  1. Figures 2-3, 5-9 (LALIGN examples), and 10-12 (binding curve examples) could be relocated to the supplementary materials.

  1. The authors' comprehensive summary of the possible mechanism for COVID-19-induced autoimmune cardiopathies is commendable. Figures 17-20 could be combined to create a highlighted figure for this paper.

  1. Please address the text shift in Table 1 and redesign the figure to ensure clarity and consistency.

  1. Table 5 contains errors with the line numbers, which need to be corrected.

  1. The current title "COVID-19 Autoimmune Cardiopathies May Result from Complementary Antigens that Mimic Cardiac Proteins Expressed by Both SARS-CoV-2 and Bacterial Co- or Super-Infections" could be simplified for easier comprehension of the study's main points.

  1. The reference formatting should be made consistent throughout the manuscript. For instance, citations 70-73 were underlined while others were not. There were also inconsistencies in the citation format as seen in 188/189.

  1. The introduction to the LALIGN and Waterman-Eggert score could be moved to the introduction/results section.

  1. The ELISA quantitative analysis visuals appear to be in draft form and need to be refined, particularly with regard to text size and formatting.

  1. Figures 2-3, 5-9 (LALIGN examples), and 10-12 (binding curve examples) could be relocated to the supplementary materials.

  1. The authors' comprehensive summary of the possible mechanism for COVID-19-induced autoimmune cardiopathies is commendable. Figures 17-20 could be combined to create a highlighted figure for this paper.

  1. Please address the text shift in Table 1 and redesign the figure to ensure clarity and consistency.

  1. Table 5 contains errors with the line numbers, which need to be corrected.

Author Response

REVEIWER 2

Open Review

( ) I would not like to sign my review report

(x) I would like to sign my review report

Quality of English Language

( ) I am not qualified to assess the quality of English in this paper

( ) English very difficult to understand/incomprehensible

( ) Extensive editing of English language required

( ) Moderate editing of English language required

(x) Minor editing of English language required

( ) English language fine. No issues detected

                Yes          Can be improved              Must be improved           Not applicable

Does the introduction provide sufficient background and include all relevant references?

                (x)           ( )            ( )            ( )

Are all the cited references relevant to the research?

                ( )            (x)           ( )            ( )

Is the research design appropriate?

                (x)           ( )            ( )            ( )

Are the methods adequately described?

                (x)           ( )            ( )            ( )

Are the results clearly presented?

                (x)           ( )            ( )            ( )

Are the conclusions supported by the results?

                ( )            (x)           ( )            ( )

Comments and Suggestions for Authors

Ijms-2522608

This intriguing study investigates whether antibodies to SARS-CoV-2 mirror cardiac antigens and if such resemblance is adequate to justify the risk of autoimmune myocarditis associated with COVID-19 and its vaccination. It also considers the potential role of bacterial co-infections, both as incidental infections or as active instigators of autoimmune myocarditis. The research reveals that some antibodies against SARS-CoV-2 proteins, especially those that target elements of the spike protein, show cross-reactivity with certain cardiac proteins. The study underscores that SARS-CoV-2 proteins mimic human cardiac proteins to a significantly larger extent than other viruses, such as Ad5, coxsackieviruses, or HCV, which have a known risk of autoimmune heart diseases, or the influenza A H1N1 virus, which doesn't. The information about adenovirus is particularly noteworthy, given that Ad5 serves as a delivery mechanism for some of the SARS-CoV-2 spike protein vaccines currently in development. The manuscript is well-composed, and the findings are clearly laid out.

THANK  YOU FOR THE POSITIVE EVALUATION!

However, some changes are suggested to enhance the manuscript's quality:

    The current title "COVID-19 Autoimmune Cardiopathies May Result from Complementary Antigens that Mimic Cardiac Proteins Expressed by Both SARS-CoV-2 and Bacterial Co- or Super-Infections" could be simplified for easier comprehension of the study's main points.

HOW ABOUT: “SARS-CoV-2 and Its Bacterial Co- or Super-Infections Synergize to Trigger COVID-19 Autoimmune Cardiopathies”

    The reference formatting should be made consistent throughout the manuscript. For instance, citations 70-73 were underlined while others were not. There were also inconsistencies in the citation format as seen in 188/189.

THANK YOU. SOME OF THESE PROBLEMS WERE EVIDENTLY INTRODUCED DURING UPLOADING OF THE MANUSCRIPT SINCE THEY ARE NOT IN OUR ORIGINAL. WE HAVE CORRECTED THE PROBLEM WITH CITATIONS 70-73 IN THIS VERSION BUT IGNORED THE OTHER FORMATTING ISSUES BECAUSE WE HAVE PUBLISHED WITH IJMS MANY TIMES AND HAVE LEARNED THAT THE EDITORIAL STAFF HAVE A PROGRAM THAT WILL CORRECT ALL REFERENCES TO THEIR PREFERRED, CORRECT FORMAT, REGARDLESS OF HOW THE REFERENCE IS ENTERED. WE THEREFORE DO NOT WORRY ABOUT REFERENCE FORMATTING. IT’S DONE AUTOMATICALLY DURING EDITING.

    The introduction to the LALIGN and Waterman-Eggert score could be moved to the introduction/results section.

GOOD IDEA! DONE. AND HIGHLIGHTED IN YELLOW FOR EASE OF IDENTIFICATION.

    The ELISA quantitative analysis visuals appear to be in draft form and need to be refined, particularly with regard to text size and formatting.

WE ASSUME THAT YOU MEAN, FOR EXAMPLE, TABLES 6 AND 7. IN A SENSE, YES, THESE ARE IN DRAFT FORM BECAUSE ONE OF THE THINGS WE HAVE LEARNED OVER MANY YEARS PUBLISHING WITH IJMS IS THAT NO MATTER HOW WE FORMAT OUR TABLES, THE EDITORS REFORMAT THEM DURING PAGE PROOFING. SO, WE HAVE PROVIDED THE SIMPLEST FORMAT FOR THE DATA, FROM WHICH THE EDITORS CAN REFORMAT THE TABLE TO THEIR LIKING. THIS IS AGAIN AN EDITORIAL ISSUE. 

    Figures 2-3, 5-9 (LALIGN examples), and 10-12 (binding curve examples) could be relocated to the supplementary materials.

THEY COULD BE, BUT WHY? THIS IS THE ACTUAL DATA UPON WHICH THE PAPER IS BASED! THE REST IS JUST SUMMARIES. WE PREFER TO LEAVE THE PRIMARY DATA IN THE TEXT SO THAT READERS CAN JUDGE THE QUALITY OF THE DATA… WE WILL DEFER TO THE EDITORS ON THIS ISSUE.

    The authors' comprehensive summary of the possible mechanism for COVID-19-induced autoimmune cardiopathies is commendable. Figures 17-20 could be combined to create a highlighted figure for this paper.

AGAIN, EXCELLENT IDEA! DONE.

    Please address the text shift in Table 1 and redesign the figure to ensure clarity and consistency.

AGAIN, AN EDITORIAL ISSUE. REALLY, THE REVIEWER NEEDS TO READ THE JOURNAL SUBMISSION GUIDELINES BEFORE MAKING ALL THIS PICKY RECOMMENDATIONS. THE FACT IS THAT ALL TABLES WILL BE RESET BY THE EDITORS DURING PROOFING, SO THEY AREN’T GOING TO LOOK LIKE THEIR CURRENT FORM NO MATTER HOW WE FORMAT THEM…

    Table 5 contains errors with the line numbers, which need to be corrected.

SORRY, BUT WE DON’T UNDERSTAND WHAT YOU MEAN HERE. WHAT ARE “LINE NUMBERS”?  THE NUMBERS IN THE RIGHT-HAND COLUMN ARE REFERENCE NUMBERS (CITATIONS), NOT LINE NUMBERS…. IF THERE IS SOME OTHER PROBLEM, WE’RE MISSING IT….

Comments on the Quality of English Language

Submission Date

08 July 2023

Date of this review

17 Jul 2023 07:27:35

© 1996-2023 MDPI (Basel, Switzerland) unless otherwise stated

Disclaimer Terms